# Molecular and evolutionary determinants for protein interaction within a class II aldolase/Adducin domain

**Marina E. Seheon**[ID][�], **Amalia S. Parra**[�], **Christopher A. Johnston**[ID]*

Department of Biology, University of New Mexico, Albuquerque, New Mexico, United States of America

☾ These authors contributed equally to this work.
* johnstca@unm.edu

## Abstract

The appearance of modular protein interaction domains represents a crucial step in the evolution of multicellularity. For example, the class II aldolase domain (ALDO$^{DOM}$) found within the Adducin gene family shares sequence and structural homology to a glycolytic aldolase enzyme found in many evolutionarily ancient phyla. ALDO$^{DOM}$ is best known for direct binding to actin filaments through a tetrameric assembly lacking catalytic activity. Molecular details for additional ALDO$^{DOM}$ interactions have not been resolved, nor have the sequence changes underlying the dramatic functional switch in the aldolase protein fold. Here we explore the molecular basis for the interaction between ALDO$^{DOM}$ of Hts (*Drosophila* Adducin) and the mitotic spindle regulator, Mud. Our results suggest a distinct mode of interaction, as conserved actin-contacting residues on the tetramer surface were found dispensable for Mud binding. Instead, we identify a critical role for the ALDO$^{DOM}$ C-terminal helix (CThelix), along with residues from the adjacent protomer that occur at a tetrameric interface conserved among domains and a subgroup of aldolase enzymes (ALDO$^{ENZ}$s). Truncation of the CThelix from bacterial ALDO$^{ENZ}$, or chimeric fusion with that from Hts, confers ALDO$^{DOM}$-like Mud binding. Sequence database analyses suggest ALDO$^{DOM}$ function may have arisen in the primitive metazoan phylum, *Placozoa*, which contains both an aldolase enzyme and domain capable of Mud binding. Finally, we identify a single, conserved arginine-to-glycine change that also permits Mud binding within the bacterial ALDO$^{ENZ}$. Our work provides molecular and evolutionary insights into the function of a conserved protein-binding domain within multicellular organisms.

## Introduction

Acquisition of new protein functions is a critical process in the evolution of multi-domain proteins found across proteomes of multicellular organisms. Such neofunctionalization can occur through gene duplication events, with subsequent sequence

**Data availability statement:** All relevant data are within the manuscript and its Supporting Information files.

**Funding:** Research reported in this publication was supported by the National Science Foundation (https://www.nsf.gov) under award number 2205405 (C.A.J). The content is solely the responsibility of the authors and does not necessarily represent the official views of the National Science Foundation. There was no additional external funding received for this study.

**Competing interests:** The authors have declared that no competing interests exist.

divergence leading to changes in globular structure, conformational dynamics, and protein-protein interaction surfaces or through co-option of existing structural domains to drive novel function [1–3]. Together, these events lead to the generation of complex functions in multicellular life such as cell signaling, cell adhesion, and differentiation. In many cases, however, very little is known about the underlying molecular mechanisms that establish newly evolved protein functions and the sequence changes leading to these functional adaptations.

One example of such functional switch is thought to have occurred in the class II aldolase protein fold. Current theories suggest this subclass of glycolytic aldolase enzymes (ALDO^ENZ), primarily restricted to bacteria and other evolutionarily ancient phyla, lost catalytic activity as it was integrated as an N-terminal aldolase domain (ALDO^DOM) within the multidomain Adducin family of cytoskeletal proteins widely conserved across multicellular life [4,5]. Adducin plays a critical role in organizing cortical spectrin-actin structures to facilitate a wide range of biological functions [6,7]. The exact roles of the ALDO^DOM remain poorly defined, however, and a molecular basis for its shift from a standalone metabolic enzyme to a modular domain in the Adducin protein family has not been rigorously addressed. Recently, cryo-EM structural analysis of the erythrocyte spectrin-actin complex revealed that the Adducin ALDO^DOM forms a tetramer that directly interacts with the barbed end of the actin filament, with additional contacts occurring via extended sequences N- and C-terminal to the core ALDO^DOM [8]. These may not be distinguishing properties of the ALDO^DOM, however, as several ALDO^ENZ structures reveal a similar tetrameric assembly that contributes to catalytic efficiency [4,9]. ALDO^ENZ has also been shown to bind F-actin, although the ubiquity of this interaction across aldolase subgroups and its functional significance are less clear [4,10]. Thus, identifying additional Adducin ALDO^DOM protein binding partners and elucidating how such interactions form remain important research pursuits.

We recently identified a direct interaction between the ALDO^DOM of the *Drosophila* Adducin protein, Hu li tai shao (Hts^ALDO), and a coiled-coil domain of the mitotic spindle regulatory protein, Mushroom body defect (Mud^CC). Functionally, Hts impacts the phase separation dynamics of Mud *in vitro* and is required for proper spindle positioning function *in vivo* [11]. Our previous work, however, did not fully resolve the molecular basis for the Hts/Mud interaction. Thus, an unresolved hypothesis we aim to address in the current work is whether the Hts/Mud complex forms in a similar manner to the Adducin/actin complex or, rather, through a distinct mode of interaction. To address this, we combine AlphaFold 3 (AF3) structural modeling, biochemical binding assays, and comparative sequence analysis of ALDO^ENZ and ALDO^DOM proteins across diverse phyla to formulate a molecular model for protein binding within the tetrameric class II aldolase fold, using the Hts/Mud complex as a model system. Unlike the Adducin/actin complex, mutation of conserved residues on the Hts tetramer surface did not impair Mud binding. In contrast, alteration of the Hts^ALDO C-terminal helix (CThelix), which AF3 modeling revealed contains an extended loop insertion compared to known bacterial enzyme structures, strongly diminished Mud binding. Mutation of conserved CThelix-contacting residues from an adjacent

protomer, which form similar contacts in Adducin and ALDO[ENZ] tetramers, also significantly reduced Mud binding. Further analysis of bacterial ALDO[ENZ]s found that truncation of their CThelix, or chimeric replacement with that from Hts, conferred strong Mud binding to these otherwise binding-deficient enzymes. A similar ALDO[ENZ] gain-of-function was identified following substitution of a single arginine, occurring at a critical tetrameric interface, to a highly conserved glycine within ALDO[DOM]s. Finally, sequence database searches across major phyla suggest that ALDO[DOM] may have first appeared within *Placozoa*, which were also found to have retained a putative class II ALDO[ENZ]. In contrast to bacterial ALDO[ENZ], the *Placozoan* enzyme showed robust Mud binding that was compromised in a chimeric protein with the bacterial CThelix. Overall, our data provide new molecular insights into the Hts/Mud complex and implicate a crucial regulatory role for the CThelix within the class II aldolase fold in the evolution of its function as a protein binding domain.

## Materials and methods

### 1. Molecular cloning

Hts and Mud sequences were amplified from a cDNA library prepared from *Drosophila* S2 cells. Additional aldolase enzyme and domain genes were amplified from plasmids constructed through complete gene synthesis by Genewiz (Azenta Life Sciences, South Plainfield, NJ, USA). All aldolase enzyme and domain sequences were cloned into the pMAL-C5x plasmid using 5'-NdeI and 3'-SalI restriction sites. The Mud[CC] domain was cloned into pBH with an N-terminal TEV-cleavable hexahistidine tag using 5'-BamHI and 3'-XhoI restriction sites.

### 2. Protein purification

All proteins were expressed in BL21(DE3) *E. coli* under induction of isopropyl β-D-1-thiogalactopyranoside (IPTG) and grown in standard Luria–Bertani broth supplemented with 100 µg/ml ampicillin. Transformed cells were grown at 37°C to an $OD_{600}$ ~0.6 and induced with 0.2 mM IPTG overnight at 18°C. Cells were harvested by centrifugation ($5000 \times g$ for 10 min), and bacterial pellets were resuspended in lysis buffer and flash-frozen in liquid nitrogen. Cells were lysed using a Branson digital sonifier and clarified by centrifugation ($12,000 \times g$ for 30 min).

For 6×His-tagged Mud[CC], cells were lysed in N1 buffer (50 mM Tris pH8, 300 mM NaCl, 10 mM imidazole) and coupled to Ni-NTA resin (Thermo Fisher Scientific, catalog #88222) for 3 h at 4°C. Following extensive washing, protein was eluted with N2 buffer (50 mM Tris pH8, 300 mM NaCl, 300 mM imidazole). The 6×His tag was removed using TEV protease during overnight dialysis into N1 buffer. Cleaved product was reverse affinity purified by a second incubation with Ni-NTA resin and collection of the unbound fraction. Final purification was carried out using an S200-sephadex size exclusion column equilibrated in storage buffer (20 mM Tris pH8, 200 mM NaCl, 2 mM DTT).

For all MBP-tagged aldolase proteins, cells were lysed in Phosphate Buffered Saline (PBS), and lysate was then clarified by centrifugation ($12,000 \times g$ for 30 min) and immediately flash frozen in liquid nitrogen for storage at −80°C. Proteins were isolated by coupling to amylose resin followed by extensive washing prior to their direct use in pulldown assays (see below).

### 3. Pulldown assays

MBP-fused aldolase proteins were absorbed to amylose agarose for 30 min at 4°C and washed three times in wash buffer (20 mM Tris, pH 8, 100 mM NaCl, 1 mM DTT, and 0.2% Triton-X100) to remove unbound protein. Subsequently, soluble Mud[CC] prey protein was added at varying concentrations for 2 h at 4°C with constant rocking in wash buffer. Incubation for different times (e.g., 1 or 3 h at 4°C, or 1 h at room temperature) produced similar results, indicating that this experimental framework had established equilibrium binding conditions. Reactions were then washed four times in wash buffer, and resolved samples were analyzed by Coomassie blue staining of SDS-PAGE gels. All gels shown in figures are representative of 3–5 independent experiments. Each independent experiment measured a single data point for indicated conditions.

Data shown in binding isotherms represent the mean standard ± deviation for measurements from the 3–5 replicates examining the specific indicated concentration and binding conditions.

All interactions were quantified using ImageJ software. Briefly, gel images were converted to greyscale and individual band intensities were measured using the boxed 'Measure' analysis tool. The size of measurement box was kept the same across all concentrations. To ensure consistent measurements of bound proteins across experimental conditions, the measurement box was set by the size of the MBP-fusion protein bait. Binding curves shown in figures plot measured band intensities (expressed as arbitrary units, 'AU') for bound Mud protein calculated by subtracting background signal quantified from the zero added Mud control in each experiment. Dissociation binding constants were calculated in Graph-Pad Prism using a one-site binding isotherm regression analysis. All plots and statistics were also performed in Prism.

## 4. Structural modeling

Protein structural modeling was primarily performed using the AlphaFold server (www.alphafoldserver.com) utilizing the current AlphaFold 3 code [12]. Primary protein sequences were input along with the desired copy number (e.g., '4' for prediction and modeling of tetrameric assemblies). Confidence metrics were interpreted for model accuracy. Monomeric structures were assessed using predicted template modeling (pTM) scores; tetrameric structures were assessed using this along with the interface predicted template modeling (ipTM) scores, both of which assess the accuracy of the entire structure [13,14]. All models presented scored above the 0.6 threshold for likelihood, with most all scoring above the 0.8 threshold for high-quality predictions. Models were further interpreted and shown images were rendered in PyMOL (www.pymol.org).

Two additional models of the Hts ALDO$^{DOM}$ tetramer were constructed for comparison and validation of the AF3 model. First, Rosetta Comparative Modeling (RosettaCM) was used to generate an Hts homology model based on the porcine Adducin heterotetramer [8,15]. Modeling was performed using the Robetta server (http://robetta.bakerlab.org), where Hts amino acids 110–401 were input for sequence alignment and subsequent modeling against the PBDid 8IAH template. The SWISS-MODEL homology modeling server (https://swissmodel.expasy.org) was used to generate the second Hts homology model, here using the *Pseudomonas syringae* aldolase enzyme homotetramer (PDBid 3OCR) as a template. Hts residues 110–401 were input, and a template search was performed. This search identified 8IAH as the top hit, followed by the 3OCR structure with a GMQE score of 0.64, which was selected as the input template.

## 5. Molecular Dynamics (MD) simulations

All MD simulations were performed by Creative Proteomics (Shirley, NY, USA) using the GROMACS MD simulation engine [16]. Simulations were performed in triplicate using a Linux GPU cluster. The AF3-generated Hts tetramer model was prepared for analysis using the PDB2PQR tool by adjusting pH to 8 to match buffer conditions of the biochemical pulldown experiments. Similarly, counter ions were added, and the system was solvated with 100 mM NaCl. Additional parameters were set for 1 bar pressure and 4°C temperature. MD simulations were performed for 200 ns using Leap-Frog integrator and results were analyzed using GROMACS analysis tools.

## 6. Sequence database analysis

Database searches were performed using BLAST (www.blast.ncbi.nlm.nih.gov). Information related to the sequences used as inputs for respective aldolase class searches can be found in Supplementary S1 Table. Class I and II enzymes are highly divergent in sequence, easily allowing distinctions of the Class I subgroup [4]. Class II enzymes were confirmed by AlphaFold modeling and structural alignment to that of the query sequence. For all sequences inferred as class IIa enzymes, identified proteins were not part of larger sequences typical of Adducin genes and were subsequently modeled using AlphaFold with a Zn$^{2+}$ ion to demonstrate predicted binding at the conserved histidine triad conserved within the

active site of bona fide class II aldolase enzyme [17]. Class IIa/adducin domains were identified as those located at the N-terminal region of proteins also containing sequence C-terminal to the aldolase sequence that most likely constitutes additional Adducin-specific function [7]. Phyla searched were selected as those representing the three kingdoms forming the tree of life along with additional ones at significant positions in animal evolution [18]. All multiple sequence alignments were performed using ClustalW (https://www.genome.jp/tools-bin/clustalw), and images shown were constructed in Snap-Gene software (https://www.snapgene.com).

## Results

### Structural modeling of Hts^ALDO domain reveals a distinct loop insertion in its CThelix

The recently determined cryo-EM structure of Adducin bound to a spectrin-actin complex found a heterotetrameric α/β-Adducin ALDO^DOM associated with the barbed actin end (PDBid: 8IAH; [8]). This tetrameric assembly, as well as folding of individual monomers, shares high homology with those of a subgroup of bacterial aldolase enzymes (e.g. PBDid: 3OCR; RMSD = 0.735 for tetramer alignment; Fig 1). To examine how the ALDO^DOM of Hts compares, we constructed a tetrameric model using AlphaFold 3 (AF3; [12]), with an input sequence of Hts amino acids (residues 110–401) that align with those that constitute the complete aldolase fold in α-Adducin [8]. *Drosophila* contain only a single Adducin gene that corresponds to α-Adducin in mammals, and thus the AF3 model generated constitutes an Hts homotetramer. This AF3 model showed a high overall confidence (ipTM = 0.87; pTM = 0.88), with a majority of positions within the 'Very high' (pIDDT > 90) confidence range apart from an extended loop within the final, 'C-terminal helix' (CThelix) of each aldolase domain (Fig 1A). This model showed high homology to both the mammalian Adducin domain and bacterial enzyme tetramers (Fig 1B,1C). Results of AF3 modeling thus demonstrate that the Hts aldolase domain is structurally conserved with that of mammalian Adducin as well as the subclass of bacterial aldolase enzymes, both in its globular fold and tetrameric assembly. This structural model will serve as the basis to further explore how the Hts aldolase domain binds its partner protein Mud and for comparison with the Adducin/actin complex.

To corroborate the AF3 structure, we generated additional Hts tetramer assemblies using homology modeling approaches. Rosetta Comparative Modeling (RosettaCM) was used to generate a model from the cryo-EM structure of porcine Adducin heterotetramer as a template [8,15]. An additional homology model was generated using SWISS-MODEL from the crystal structure of *Pseudomonas syringae* Aldolase enzyme homotetramer as a template [PDBid 3OCR; [19]]. Both homology models showed strong overall similarity to the AF3-generated structure (RMSD = 1.179 and 1.507, respectively; S1A, S1C Fig). However, the models showed divergent conformations of the extended CThelix loop, consistent with the lower confidence predictions within this region and its potentially flexible nature (see further discussion below).

We also performed molecular dynamics (MD) simulation analysis to evaluate the stability to the AF3 structure. Triplicate simulations each demonstrated that the tetramer is stable, with all four subunits remaining intact throughout the simulation (S2 Fig). RMSD for the Ca backbone equilibrated quickly and stabilized at an average 3–3.5 Å over the 200 ns simulation (S2D Fig). Radius of gyration (Rg) values remained stable between 3.24–3.26 nm, as did solvent accessible surface area (SASA) values (S2E-S2F Fig). Comparison of the three simulated structures with the AF3 model revealed overall alignment RMSDs of 1.646, 1.486, and 1.769 Å (S2A-S2C Fig). Alterations in the CThelix conformation, particularly the extended loop insert, were observable in each simulated structure suggesting a higher intrinsic flexibility of this region (S2A-S2C Fig). Overall, from these combined modeling and simulation approaches, we conclude that the Hts ALDO^DOM forms a tetrameric assembly with high structural homology to related aldolase fold proteins.

As mentioned, each Hts domain monomer has a large, conspicuous loop within its CThelix (Fig 1). Alignment of single monomers demonstrated that Adducin has a smaller loop insertion, and the bacterial enzyme has no significant loop but instead a short break in an otherwise contiguous CThelix (Fig 1B-1D). AF3 modeling of the bacterial enzyme was able to accurately place the active site catalytic $Zn^{2+}$ ion (Fig 1D). In contrast, neither Adducin nor Hts modeling showed such

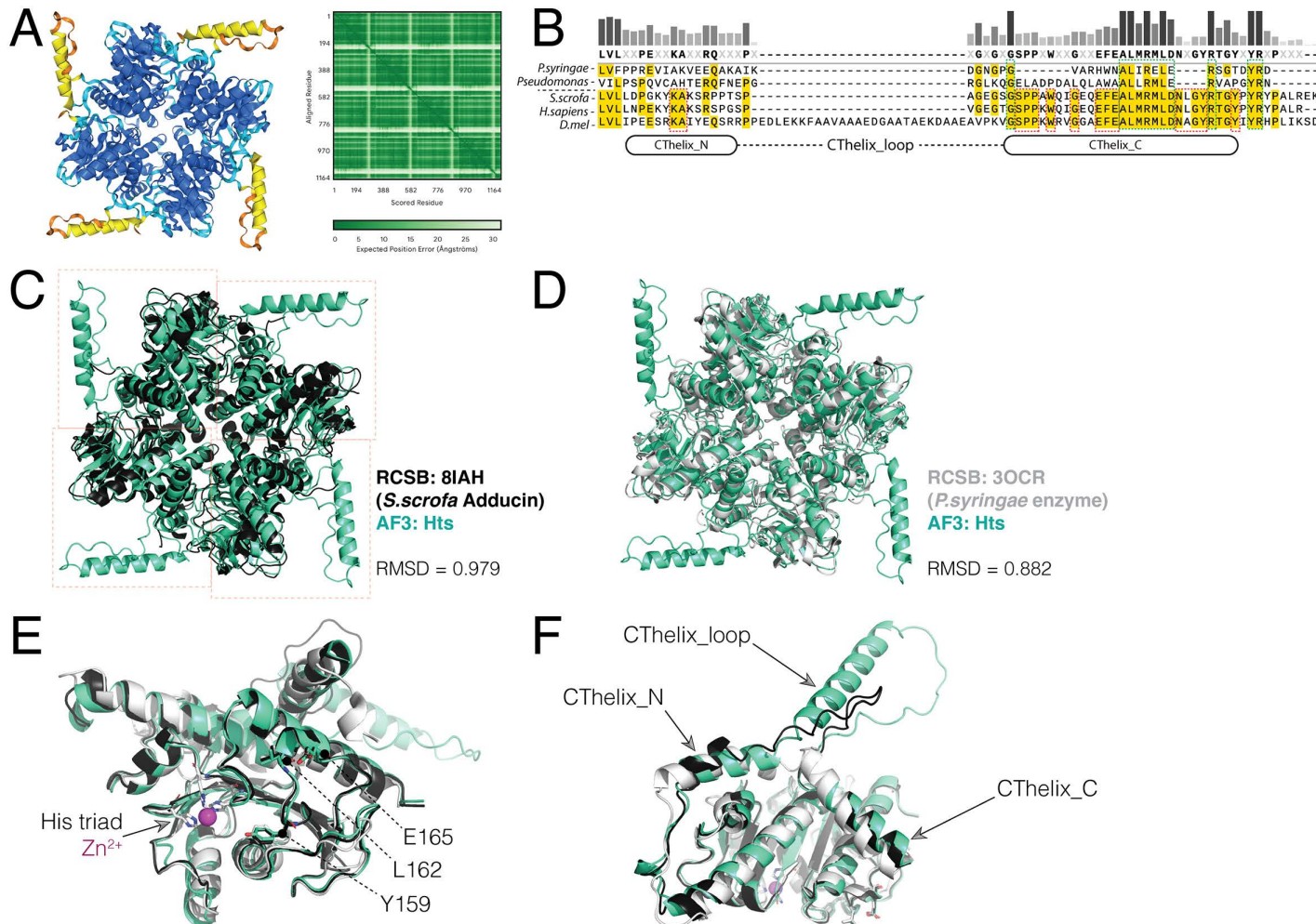

**Fig 1. AlphaFold modeling identifies an extended CThelix loop in a tetrameric Hts aldolase domain assembly. (A):** *Left*: AF3 modeling output results for the Hts tetramer color coded by per-atom confidence estimate (pLDDT) on 0-100 scale (*Blue* – Very high: pLDDT > 90, *Cyan* – Confident: 90 > pLDDT > 70, *Yellow* – Low: 70 > pLDDT > 50, *Orange* – Very low: pLDDT < 50). The core aldolase fold in each of the four Hts monomers contains a majority of 'Very high' scoring positions, with a minor fraction of 'Confident'. In contrast, the extended CThelix loop showed scores in the 'Low' and 'Very low' categories. *Right*: Predicted alignment error plot shows the expected positional error for each residue, also indicating a high confidence level (dark green) in model accuracy apart from the CThelix loop positions (light green). **(B):** Multiple sequence alignment of CThelix sequences from bacterial aldolase enzymes (*P. syringae* and *Pseudomonas*) and aldolase domains from pig (*Sus scrofa, S.scrofa*), human (*Homo sapiens, H.sapiens*), and fruit fly (*Drosophila melanogaster, D.mel*) demonstrates the extended insert sequence in fly Hts domain that occurs between the N- and C-terminal regions of the CThelix. Also indicated are positions conserved among enzymes and domains (*dashed green boxes*) and those only conserved among domains (*dashed red boxes*). **(C):** Overlay of fly Hts (*bluegreen*) and porcine Adducin (*black*) tetramers shows a high degree of similarities, with an RMSD = 0.979, calculated from a superimposition alignment in PyMOL. **(D):** Overlay of fly Hts (*bluegreen*) and bacterial aldolase enzyme (*white*) tetramers shows a high degree of similarities, with an RMSD = 0.882. **(E):** Overlay of Hts, Adducin, and aldolase enzyme monomers indicating conserved YLE motif positions (numbered according to fly Hts) that are positioned adjacent to the bacterial enzyme active site containing the histidine triad ('His triad'; *white sticks*) and catalytic $Zn^{2+}$ (*magenta*). **(F):** Overlay of Hts, Adducin, and aldolase enzyme monomers demonstrating the conserved structure of the N- and C-terminal segments of the CThelix ('CThelix_N' and 'CThelix_C', respectively) as well as the divergent loop insert segment ('CThelix_loop'). Fly Hts contains a large CThelix loop, Adducin shows a smaller insert, and the bacterial enzyme lacks this segment entirely.

placement, consistent with divergence in the catalytic site and loss of enzymatic function in transition to aldolase domains [7]. Overall, these results demonstrate the conserved nature of the core aldolase fold, highlight a marked difference in the CThelix, and substantiate the accuracy of AF3 modeling for both enzymes and domains.

## The CThelix loop regulates Mud^CC direct interaction with the Hts^ALDO domain

Having identified the CThelix as the most structurally divergent element in aldolase proteins, we next inspected the role for this Hts sequence in regulating Mud binding. To do so, we immobilized Maltose-binding protein (MBP)-fused Hts proteins on agarose resin and co-incubated with purified Mud^CC domain, which is responsible for direct Hts interaction (Fig 2A). The full-length Hts ALDO^DOM strongly bound Mud with low-micromolar affinity, similar to our previous studies (Fig 2B; [11]). Deletion of the large CThelix loop insertion resulted in a significant loss of Mud binding, as quantified by a reduction in the maximal binding ($B_{max}$), with minimal effects on the affinity of this diminished interaction efficacy (Fig 2C). Note that for a complete list of mutants tested in this study, along with their evolutionary significance and functional impact, see Table 1. This result suggests that Mud binding could occur directly within this loop insertion. However, truncation of the entire CThelix, including the loop insertion, surprisingly restored complete Mud binding comparable to the full-length ALDO^DOM (Fig 2D), indicating that the binding site occurs within the minimal aldolase core fold. Taken together (Fig 2E), these data suggest a possible regulatory role for the CThelix in Mud binding, with the loop insertion playing an important but indirect part of the interaction.

## Mutations at conserved surface residues on the Hts tetramer do not impair Mud binding

We next sought to identify a specific Mud binding site within the Hts aldolase domain. The ability of Mud to bind following removal of the Hts CThelix narrows the binding site to a minimal aldolase fold (residues 111–310), eliminating the small number of conserved contacts between the Adducin CThelix and actin as likely contributors. The CThelix loop insertion is notably disordered in the Adducin/Actin complex [8], consistent with it not being directly involved similar to our results above with Hts/Mud binding. Furthermore, a majority of Adducin contacts with actin involve residues in extended sequences N- and C-terminal to the minimal aldolase core [8], with D339 (D310 in Hts) representing one exception. These findings together suggest that Mud binding occurs at a unique site within Hts. To examine this directly, we constructed a single alanine mutant at the conserved D310, along with three additional double mutants that were identified as conserved residues on surface-exposed regions of the Hts core tetramer (Q151A/E152A, Q177A/V181A, R205A/D207A; Fig 3A). Saturation binding experiments found that none of these mutants impaired Mud binding relative to a wild-type control Hts (Fig 3B-3C). These data suggest that Mud interacts with a distinct site within the Hts ALDO^DOM that lies outside of conserved regions of the tetrameric complex surface.

## CThelix deletion confers Mud binding to an ancient class II aldolase enzyme

We next explored Mud binding to a primitive aldolase enzyme from *Pseudomonas syringae*, whose CThelix lacks the loop insertion found in Hts and α-Adducin domains but nevertheless makes similar contacts to its adjacent protomer within an otherwise conserved tetrameric assembly (Fig 1D and 4A). This ALDO^ENZ lacked significant Mud binding, a result not unexpected considering its suspected role as a metabolic enzyme (Fig 4B-4C). Remarkably, deletion of the CThelix from this ALDO^ENZ conferred robust Mud binding (Fig 4B-4C), which was nearly equivalent to that seen with the ALDO^DOM of Hts, both 'full length' and ΔCThelix. AF3 modeling indicated that CThelix removal from the *Pseudomonas syringae* enzyme does not prevent tetramer assembly (Fig 4A), consistent with the known tetramerization of aldolase enzymes naturally lacking a CThelix (see below). To corroborate this finding, we performed similar experiments with the homologous aldolase enzyme from an alternative *Pseudomonas* species (labeled simply '*Pseudomonas*' throughout). We found this enzyme also lacked Mud binding in its full-length, wild-type form, but showed a robust interaction following deletion of its CThelix similar to the *Pseudomonas syringae* aldolase (Fig 4D).

## Chimeric CThelix aldolase proteins display differential Mud binding dynamics

We next constructed two chimeric aldolase proteins, swapping the CThelices between the Hts ALDO^DOM and the *Pseudomonas syringae* ALDO^ENZ, to further investigate the impact of this divergent structural element in Mud binding (Fig 5A).

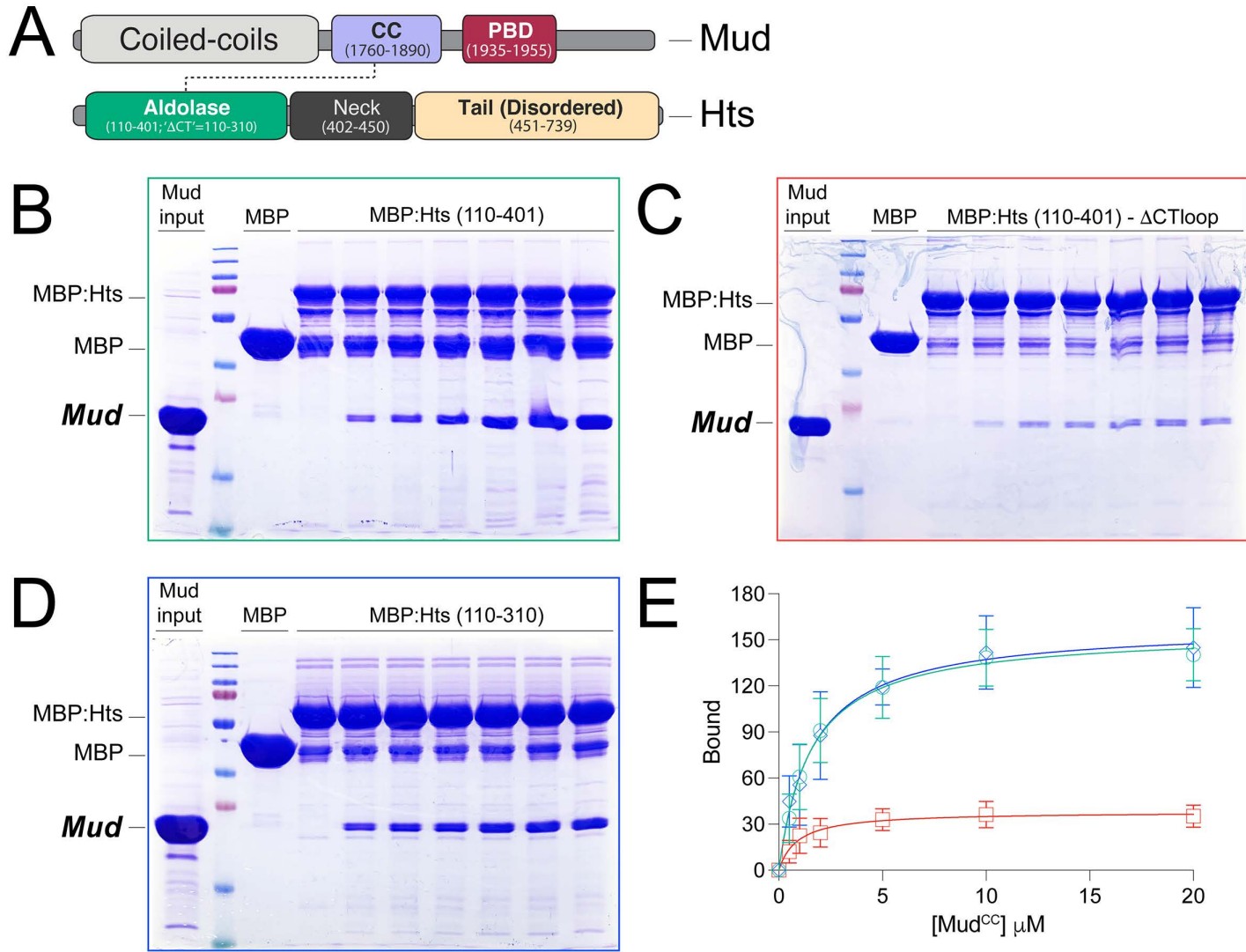

**Fig 2. The CThelix regulates direct interaction between Hts aldolase domain and Mud$^{CC}$. (A):** Domain diagrams for Mud and Hts proteins are shown (not to scale). Mud (*top*) contains extended coiled coils at the N-terminus (*grey*), followed by a shorter, C-terminal coiled coil (CC, *purple*) that is the focus of the current study. This Mud$^{CC}$ domain is followed by the Pins-binding domain (PBD, *red*). Hts (*bottom*) contains the N-terminal Aldolase domain (*bluegreen*) that is also the focus of this study, followed by the Neck (*dark grey*) and disordered Tail (*light orange*) regions. **(B):** Representative gel image depicting interaction between the full-length Hts aldolase domain and Mud$^{CC}$. MBP-fused Hts [MBP:Hts (110-401)] or MBP alone control were incubated in the absence or presence of increasing concentrations of Mud$^{CC}$. Gel shown is representative of 4 independent experiments. **(C):** Representative gel image depicting interaction between a CThelix loop deletion in the Hts aldolase domain and Mud$^{CC}$. MBP-fused Hts [MBP:Hts (110-401)-·CTloop] or MBP alone control were incubated in the absence or presence of increasing concentrations of Mud$^{CC}$. Gel shown is representative of 4 independent experiments. **(D):** Representative gel image depicting interaction between a CThelix truncation in Hts aldolase domain and Mud$^{CC}$. MBP-fused Hts [MBP:Hts (110-310)] or MBP alone control were incubated in the absence or presence of increasing concentrations of Mud$^{CC}$. Gel shown is representative of 4 independent experiments. **(E):** Saturation binding curves for all the above experiments demonstrate the effects of CTloop deletion and CThelix truncation mutants on Mud$^{CC}$ binding to the Hts aldolase domain. Full-length Hts domain (*green*) shows a robust, dose-dependent binding. Deletion of the CTloop (*red*) significantly impairs Mud binding, whereas truncation of the entire CThelix (*blue*) restores strong binding indistinguishable from the full-length domain.

**Table 1. List of key mutants examined together with their evolutionary significance and functional impact.**

| Mutation | Evolutionary significance | Functional impact |
|---|---|---|
| Hts_ΔCTloop (deletes amino acids 331–352 from Hts (110–401) | This deletion mutant removes the large loop insert identified in the Hts aldolase domain CThelix. Bacterial aldolase enzymes examined lack this large loop segment. | Significantly reduces the maximal Mud binding of the Hts aldolase domain (see *Fig 2C*). |
| Hts_ΔCThelix (deletes amino acids 311–401 from Hts (110–401), yielding the 'core' aldolase fold (amino acids 110–310) | This deletion mutant removes the entire CThelix from the Hts aldolase domain. The CThelix sequence is divergent in aldolase enzymes and domains studied herein. A subgroup of aldolase enzymes (the fuculose aldolases), that do not bind Mud, lack this sequence entirely. | Restores Mud binding compared to the CTloop deletion mutant. This mutant binds Mud equivalent to the full Hts domain (see *Fig 2D*). Together with the CTloop deletion, demonstrates a regulatory role for the CThelix in Mud binding to the Hts aldolase domain. Indicates the Mud binding site lies within the core aldolase domain fold. |
| Hts_Q151A/E152A, Hts_Q177A/V181A, and Hts_R025A/D207A | These three double alanine mutants occur at amino acids conserved among Adducin aldolase domains that are also surface exposed on the tetrameric assembly. These are designed to test for a possible Mud binding site at conserved patches on the tetramer surface. | No significant loss of Mud binding was measured, suggesting Mud binding does not occur at these tetramer surfaces. Further suggests the Mud binding site may lie at a non-surface exposed site (see *Fig 3*). |
| Hts_D310A | This single alanine mutant is to a conserved aspartic acid that directly contacts F-actin in the cryo-EM structure of *S. scrofa* Adducin complex (*PDBid: 8IAH*). | No significant loss of Mud binding was measured, indicating this actin-binding residue is dispensable for Mud binding (see *Fig 3*). |
| 3OCR_ΔCT (deletes amino acids 227–271) from the *P. syringae* aldolase enzyme | This deletion mutant removes the entire CThelix from the *P. syringae* aldolase enzyme and is designed to test the hypothesis that Mud binding is restricted in bacterial aldolase enzymes by the divergent CThelix compared to the Hts aldolase domain. | Compared with the negligible binding to the full-length enzyme, this CThelix deletion mutant showed a strong gain-of-function Mud binding, which was similar to that seen in the Hts aldolase domain. Thus, removal of the CThelix permits Mud binding to this ancient aldolase enzyme (see *Fig 4C*). |
| *Pseudomonas*_ΔCT (deletes amino acids 198–251) from the *Pseudomonas* aldolase enzyme | This deletion mutant removes the entire CThelix from the *Pseudomonas* aldolase enzyme, similar to that described for the *P. syringae* aldolase above. | Similar gain-of-function Mud binding was seen in this bacterial enzyme compared with *P. syringae* above. Further demonstrates a key role for the CThelix in controlling Mud binding to aldolase fold proteins (see *Fig 4D*). |
| ENZ/Hts^CT chimera | This chimera fuses the core *P. syringae* enzyme (amino acids 1–226) with the full CThelix from Hts domain (amino acids 311–401). | This chimera shows a gain-of-function Mud binding similar to that seen with deletion of the natural CThelix from bacterial aldolase enzymes. It demonstrates that the core enzyme fold can bind Mud when a competent CThelix is present (see *Fig 5*). |
| Hts/ENZ^CT chimera | This chimera fuses the core Hts domain (amino acids 110–310) with the full CThelix from *P. syringae* enzyme (amino acids 227–271). | This chimera shows a modest reduction in Mud binding compared to Hts^ALDO. It demonstrates that the ability to bind Mud can be reduced when a restrictive CThelix is present (see *Fig 5*). |
| Hts_YLEaaa (Y159A/L162A/E165A), and Hts_L139A | These alanine mutants are designed at amino acids that [1] show strong conservation between Mud-binding aldolase enzymes and domains, and [2] make direct contacts with the CThelix of neighboring protomers in their respective tetrameric assemblies. | Both mutants significantly impair Mud binding compared to wild-type, suggesting these conserved residues are critical for Mud interaction (see *Fig 7A*). |
| Hts_HTaa (H164A/T167A), and Hts_R128A | These alanine mutants are also designed at amino acids that show conservation between Mud-binding aldolase enzymes and domains. These are adjacent to those listed above in tetrameric structures. | Neither mutant impaired Mud binding compared to wild-type, suggesting these conserved residues are not critical for Mud interaction (see *Fig 7B*). |
| 3OCR_FLEaaa (F72A/L75A/E78A) | This triple alanine mutant is designed at *P. syringae* residues analogous to the YLE motif identified in Hts above. | When examined in the background of the gain-of-function ·CThelix *P. syringae* aldolase enzyme, this mutant strongly reduces Mud binding. This suggests a common Mud binding site within the core aldolase fold (see *Fig 7C*). |
| 3OCR_R47G | This single missense mutant replaces the conserved *P. syringae* arginine residue identified in primitive aldolase enzymes with the conserved glycine found in some enzymes and all aldolase domains examined. This residue is located at a key tetrameric assembly interface. | The R47G mutant shows a strong gain-of-function Mud binding compared to wild-type enzyme. Mud binding to R47G is similar to that seen with ·CThelix and CThelix chimeric enzymes, as well as the Hts domain. This indicates that a single amino acid substitution can confer new function within the aldolase fold (see *Fig 11*). |
| Hts_G134R | Reciprocal to that described above, this single missense mutant replaces the conserved Hts glycine residue with arginine to mimic primitive aldolase enzyme sequence at this tetrameric interface site. | The G134R mutant shows a modest reduction in Mud binding affinity compared to wild-type Hts domain. This indicates that a strong reversion of domain function cannot be achieved by a single mutation (see *Fig 11*). |

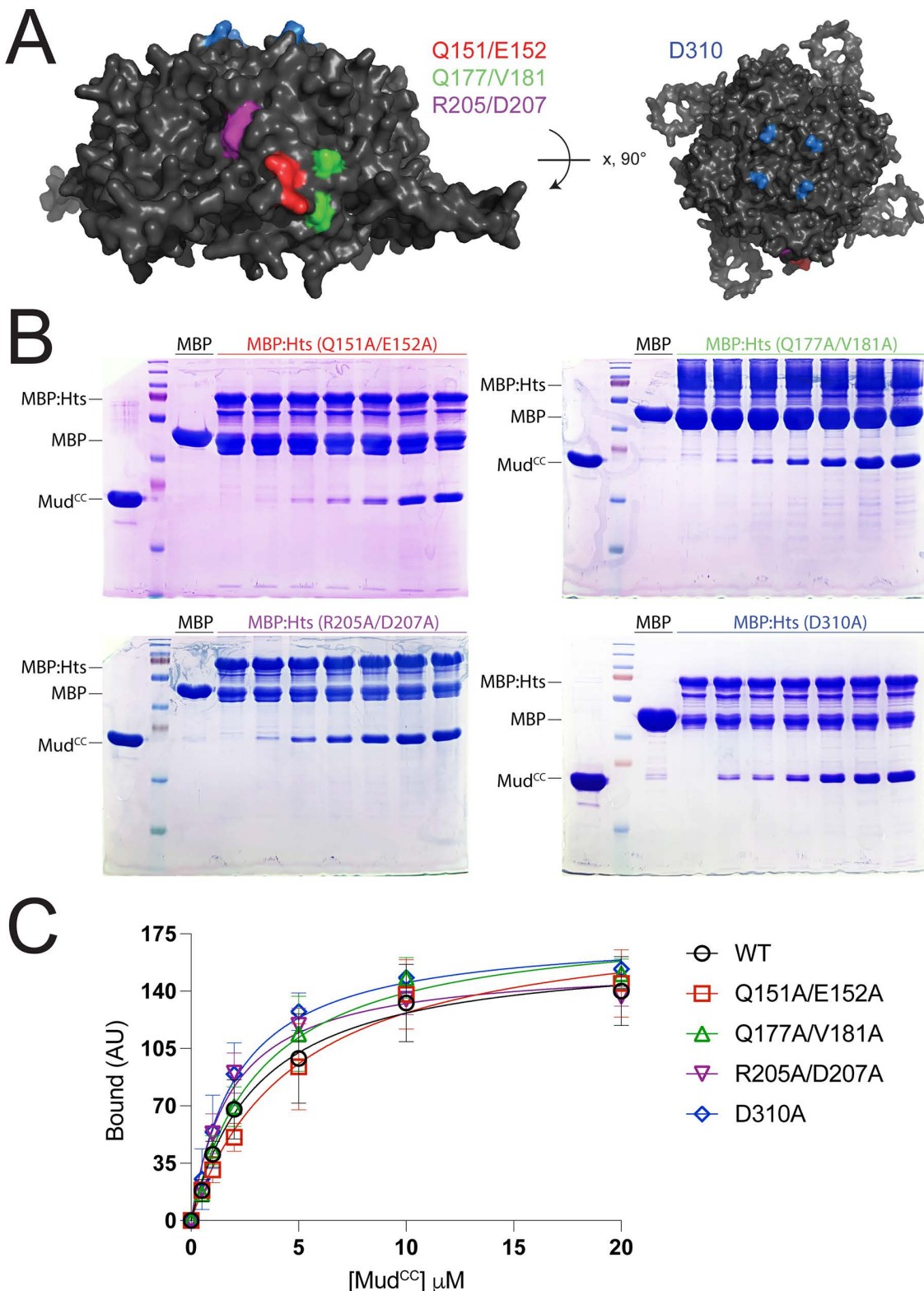

**Fig 3. Conserved surface exposed Hts residues do not contribute to Mud binding. (A):** *Left*: Lateral view of Hts tetramer highlighting three conserved surfaces residue pairs targeted for mutational analysis. *Right*: Rotation depicting the tetramer face with the conserved actin-contacting D310

residue also targeted for mutational analysis. **(B)**: Representative gel images depicting interaction between in the Hts aldolase domain mutants and Mud$^{CC}$. Each indicated MBP-fused Hts mutant or MBP alone control were incubated in the absence or presence of increasing concentrations of Mud$^{CC}$. Gels shown are each representative of 3-5 independent experiments. **(C)**: Saturation binding curves for Hts aldolase domain wild-type and mutants binding Mud$^{CC}$. All mutants showed similar binding dynamics compared to control Hts.

Compared to the wild-type enzyme that lacks significant Mud binding, chimeric fusion with the Hts CThelix resulted in a strong interaction similar to that seen following deletion of the enzyme CThelix (Fig 5B). In contrast, fusion of the Hts domain with the CThelix from the bacterial enzyme reduced Mud binding affinity compared to the wild-type domain (Fig 5C). However, this inhibitory effect of the enzyme CThelix on Hts function was weaker compared to the strong gain-of-function seen in the reciprocal chimera. Taken together, these results further highlight a key regulatory role for the CThelix in controlling the Mud binding activity within the aldolase fold.

### Identification of a conserved, CThelix-binding 'YLE motif' within the core aldolase fold

Considering the gain-of-function Mud binding following truncation or chimerism of the aldolase enzyme CThelix, we next considered that the binding site may occur in a location repressed by this sequence in the full-length enzyme. To explore this idea, we constructed a multiple sequence alignment of Hts with several aldolase enzymes and Adducin domains to identify conserved residues near protomer interfaces that contact the CThelix in tetrameric structures (Fig 6). This analysis revealed several such positions within interconnecting loops of the core aldolase fold that directly contact the CThelix of the neighboring protomer (Fig 6A). Close inspection of this interface shows a particularly remarkable conservation within a triad of residues (F88/L91/E94 in the *Pseudomonas syringae* enzyme and Y159/L162/E165 in the Hts domain; termed the 'YLE motif' herein) that contact a series of CThelix residues, some of which display similar conservation along with additional residues that constitute unique contacts in the aldolase domain (Fig 6C). The RosettaCM and SWISS-MODEL homology models both showed a similar disposition of the YLE motif residues and their critical contacts in the CThelix compared to the AF3-generated structure (Fig S1B,D).

### The 'YLE motif' defines a putative Mud binding site

To determine the role of these conserved positions in the Hts ALDO$^{DOM}$, we constructed a series of alanine mutants and measured their Mud binding capacity relative to wild-type. Triple mutation of the YLE motif resulted in a complete loss of Mud binding when examined within the context of either the full-length domain or the minimal core (e.g., ΔCThelix; Fig 7A). A single L139A also significantly impaired binding (Fig 7A). In contrast, double mutation of H164/T167 bound Mud similar to wild-type, as did a single R128A mutant (Fig 7B). These data highlight a critical role specifically for the YLE motif in Mud binding, while discounting a role for the adjacent H164/T167 and R128 residues. To further examine the importance of the YLE motif, we tested the effects of mutations to the equivalent FLE motif in the *Pseudomonas syringae* aldolase enzyme. When examined within the context of the gain-of-function ΔCThelix enzyme, this mutation resulted in significantly impaired Mud interaction, similar to the YLE mutation in Hts (Fig 7C). Overall, these data implicate an important role for the conserved YLE motif in Mud binding (Fig 7D).

### Mud binding varies widely across aldolase isozymes

Having identified a putative Mud interaction site conserved between the Hts aldolase domain and *Pseudomonas syringae* enzyme, we next widened our analysis to additional aldolase enzymes. Searching structural and sequence databases, we noted three subgroups within the Adducin-like class II aldolase enzymes (those with structural homology to the aldolase domain, referred to as 'class IIa' herein), which could be distinguished by the length of their CThelix (Fig 8A-8B). These enzymes mostly belong to a group of tetrameric L-fuculose-1-phosphate aldolases found in many pathogenic bacteria

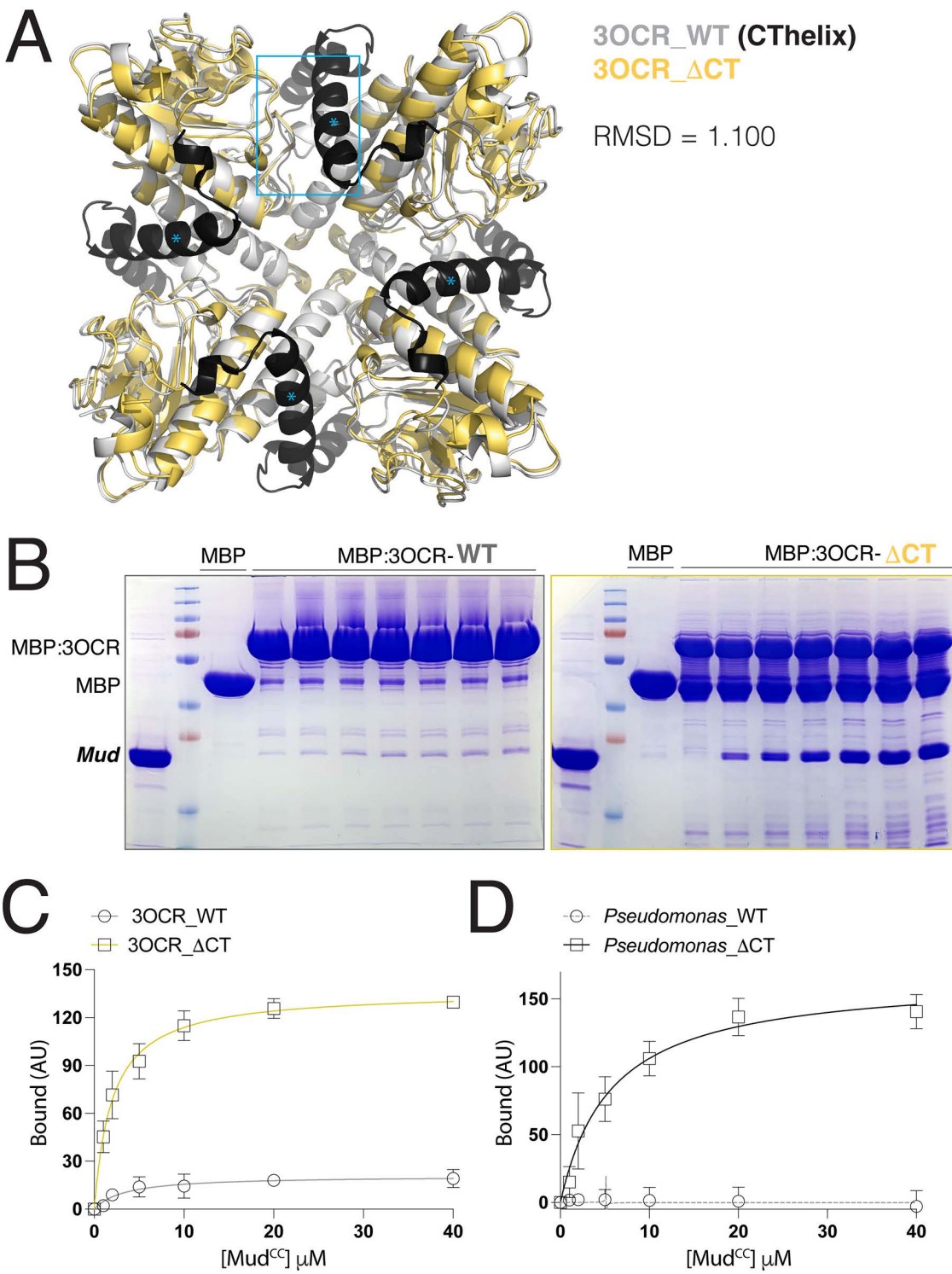

**Fig 4. CThelix deletion confers Mud binding function to a bacterial aldolase enzyme. (A):** Overlay of the full-length *P. syringae* aldolase enzyme (PDBid: 3OCR; *white* with CThelix colored *black*) tetramer with that of an AF3-generated model following deletion of the CThelix (ΔCThelix; *yellow*). The ·CThelix enzyme retains a confident prediction for tetrameric assembly (pTM score = 0.7; ipTM score = 0.62) and aligns well to the full-length enzyme structure (RMSD = 1.1). Cyan box and asterisks depict interaction of the CThelix (*) with loops in the aldolase core of its adjacent protomer. **(B):** Representative gel images depicting Mud$^{CC}$ interaction with full-length (*left*) and ΔCThelix (*right*) aldolase enzyme from *P. syringae*. Each indicated MBP-fused

aldolase enzyme or MBP alone control were incubated in the absence or presence of increasing concentrations of Mud[CC]. Gels shown are representative of 3-5 independent experiments. **(C)**: Saturation binding curves for conditions described in **(B)**. Whereas the full-length aldolase enzyme shows only minimal Mud-binding capacity, truncation of the CThelix results in a robust Mud interaction. **(D)**: Similar experiments to those described in (C) were performed using the class II aldolase enzyme from an alternative *Pseudomonas* species. No apparent Mud binding was observed with the wild-type, full length enzyme; however, deletion of the CThelix also conferred strong Mud binding to this bacterial enzyme.

[20]. The first subgroup completely lacks a CThelix (termed 'no CT' herein); nevertheless, they share a core structure and tetrameric assembly with those aldolases already discussed (e.g. PBDid: 2IRP; Fig 8B). Ostensibly, the 2IRP enzyme mimics the *Pseudomonas syringae* enzyme following truncation of its CThelix. In contrast to this enzyme, however, we found that the 'no CT' enzyme was completely incapable of binding Mud (Fig 8C). Primary sequence comparison revealed that the 'no CT' enzyme is highly divergent in the core loop sequences, including the YLE motif (Fig 8A). Additional 'no CT' aldolase structures also demonstrate that, unlike 'full CT' enzymes, the divergent residues corresponding to the YLE motif in these 'no CT' enzymes do not contact neighboring protomers in what are otherwise similar tetrameric assemblies [21]. Mutation of these corresponding residues to YLE in the 2IRP enzyme did not lead to a gain of Mud binding function (Fig 8C), indicating this motif is necessary for Mud binding in relevant aldolase proteins but not sufficient for those naturally lacking this activity.

The second enzyme subgroup identified contains a short CThelix sequence (termed 'half CT' herein), which is structurally analogous to the sequence found prior to the loop insertion in Hts (Fig 8B). Structural studies have determined a similar tetrameric assembly in this enzyme subgroup as well [e.g. PDBid: 2FUA; [22]]. As with the 'no CT' enzyme, we found that the 'half CT' enzyme was completely devoid of Mud binding. Furthermore, truncation of the shortened CThelix did not permit Mud binding (Fig 8C). Notably, as with 'no CT' enzymes, the 'half CT' aldolases also lacked conservation of the YLE motif (Fig 8A).

Lastly, database searches identified a third subgroup of fuculose aldolase enzymes, distributed across diverse fungi, that contain a CThelix of similar length to that of *Pseudomonas syringae* enzyme and Hts/Adducin domain (e.g. PDBid: 4XXF; termed 'full CT' herein; Fig 8B). This 'full CT' fuculose aldolase showed a moderate Mud binding activity that was improved upon truncation of its CThelix, although both responses were weaker than that measured with Hts (Fig 8C). Examination of its primary sequence found a partially conserved YLE motif (Fig 8A). Overall, these analyses further substantiate the CThelix and YLE motif as key molecular determinants for Mud binding in the aldolase protein fold.

### Aldolase gene expression suggests a potential Placozoan emergence of the Adducin family

Aldolases represent a large family of surprisingly diverse enzymes with divergent sequences, structures, and reaction mechanisms that nonetheless contribute similar metabolic functions through convergent evolution [4]. The starkest of these contrasts distinguish the class I and II subfamilies. Class I aldolases adopt a classic (and eponymous) TIM barrel fold and use a lysine-dependent Schiff base reaction mechanism [23,24]. Class II enzymes, in contrast, use a metal (most often $Zn^{2+}$) catalyzed reaction scheme, with a conserved histidine triad functioning to coordinate this catalytic ion [17,25]. Class II enzymes can be further categorized structurally, with one subset also adopting a TIM barrel fold and a second showing high homology to the aldolase domain of Adducin proteins, along with other subgroups of sugar lyases (e.g., 'class IIa' enzymes; [26–28]). As presented above, class IIa enzymes have a similar core structural fold but diverge in the length of their CThelices.

Considering these sequence and structural distinctions across the aldolase enzyme family, as well as the dramatic functional divergence between class IIa enzymes and the Adducin aldolase domain, we next extended our search of the protein sequence database (https://blast.ncbi.nlm.nih.gov/Blast.cgi) to identify expression patterns of these unique aldolase proteins across major phyla in an effort to further explore a potential pathway of functional evolution within this protein fold (Fig 9). We searched selected phyla for class I enzyme, class II enzyme, class IIa enzyme ('no CT', 'half CT', and 'full

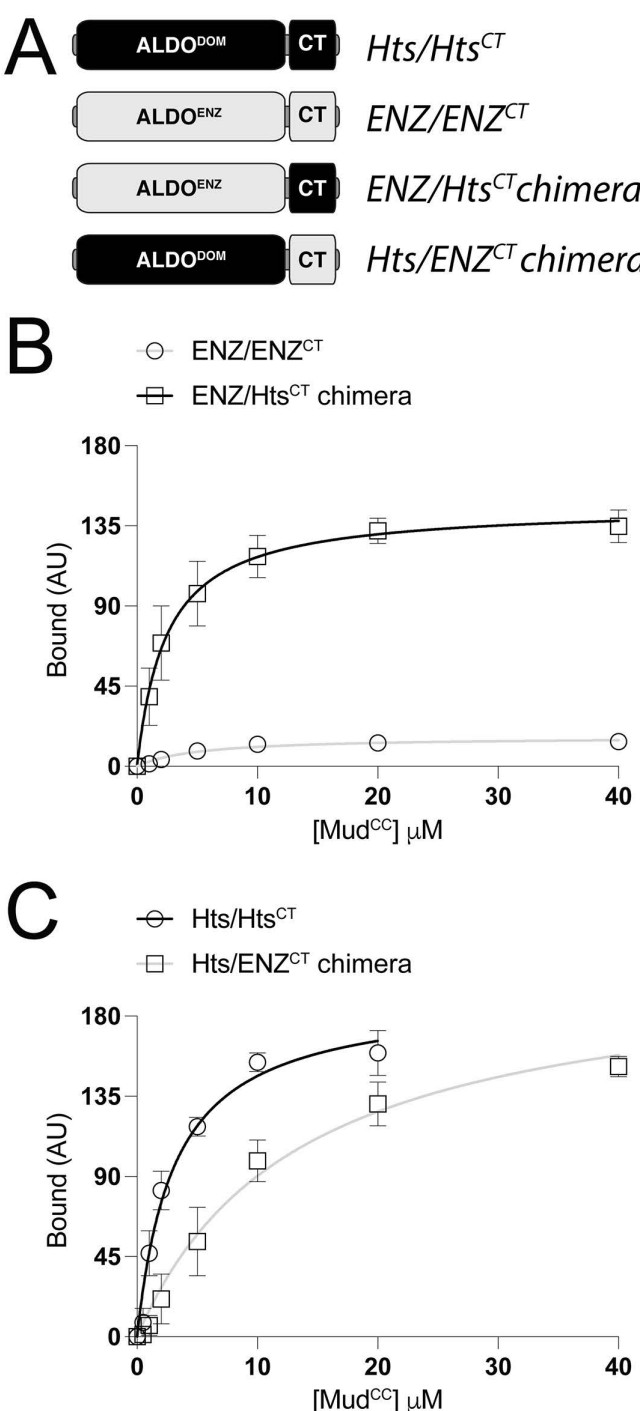

**Fig 5. Chimeric aldolase proteins support a key regulatory role for the CThelix in Mud binding function. (A):** Schematic representation of chimeric aldolase proteins shows strategy to swap CThelices ('CT') between the *Pseudomonas syringae* aldolase enzyme (*light grey*; ALDO^ENZ) and the Hts aldolase domain (*black*; ALDO^DOM). **(B):** Saturation binding curves for the full-length aldolase enzyme (ENZ/ENZ^CT; *grey circles*) and its chimeric version with the Hts CThelix (ENZ/Hts^CT; *black squares*). Whereas the aldolase enzyme shows only minimal Mud-binding capacity, the chimeric aldolase shows a robust Mud interaction. **(C):** Saturation binding curves for the Hts aldolase domain (Hts/Hts^CT; *black circles*) and its chimeric version with the aldolase enzyme CThelix (Hts/ENZ^CT; *grey squares*). The chimeric aldolase shows a moderately impaired Mud binding compared to the full-length Hts domain.

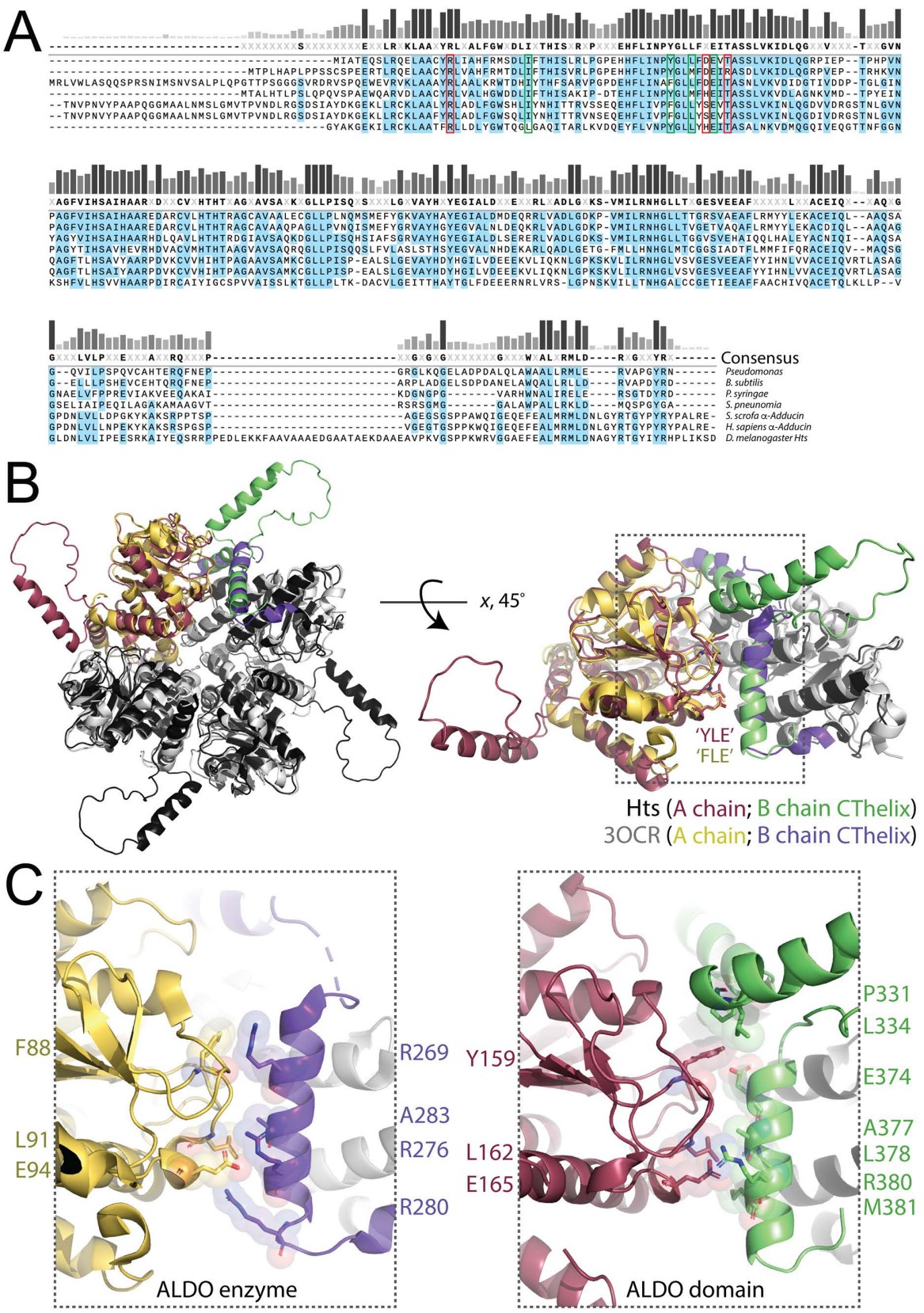

**Fig 6. Hts shares a conserved YLE motif with primitive aldolase enzymes that contacts the adjacent protomer CThelix in the tetrameric assembly. (A):** Multiple sequence alignment among several bacterial aldolase enzymes and aldolase domains. *Green boxes* indicate conserved residues in the YLE motif, whereas *red boxes* indicate additional conserved residues that neighbor the YLE motif in the core aldolase structure. These residues were tested for contributions to Mud binding (see Fig 7). **(B):** *Left*: face view tetrameric assembly overlay for Hts aldolase domain (*black*; chain A is *red*, chain B CThelix is *green*) and the *Pseudomonas syringae* aldolase enzyme (*white*; chain A is *yellow*, chain B CThelix is *purple*). *Right*: lateral view of contacts between chain A and B protomers with the conserved, CThelix-contacting Y/F-LE motif residues shown as sticks. **(C):** Zoomed views of Y/F-LE/CThelix contacts in the aldolase enzyme (*left*; ALDO enzyme) and aldolase domain (*right*; ALDO domain) demonstrate the conserved nature of these interactions. Specific contacting residues are indicated.

CT' versions), and Adducin-like aldolase domain sequences as depicted in Fig 9A. We found class I enzyme sequences present across all phyla apart from the *Choanoflagellates* (Fig 9B). Our searches also indicated that class I enzymes represents the lone subtype found across higher eukaryotes (with the exception of a class II sequence also identified in the protostome *Lophotrochozoa*), consistent with previous studies finding a loss of class II genes in these phyla [29–31]. Class II enzymes were found in bacteria and most of lower eukaryotic phyla, typically together with class I sequences (Fig 9B). Class IIa enzymes were identified among similar phyla, albeit slightly more restricted than the class II sequences (Fig 9B). Searches for the aldolase domain found a restricted pattern of expression to metazoan phyla (Fig 9B). Based on these search results, the aldolase domain appears to have emerged in *Placozoa* (or possibly *Porifera*), which were also found to be the last phylum with apparent class IIa enzyme sequences, with both a 'no CT' and 'full CT' sequence identified (Fig 9B). Notably, searches of several *Choanoflagellate* species (an outgroup of metazoa), which have been shown to be a critical origin for other multi-domain scaffold proteins similar to Adducin [32–36], revealed neither class IIa enzymes nor aldolase domain-containing Adducin sequences (Fig 9B). Overall, our sequence analysis supports existing theories of the aldolase enzyme family evolution and highlight a potentially important role for *Placozoa*, a phylum of basal animals, in the class IIa and Adducin gene subgroups. This phylum of marine invertebrates forms aggregate 'blobs' of cells and are thought to represent one of the simplest forms of animal life. As such, they represent a critical position in the move to complex, multicellular life that could reveal essential molecular insights into this evolutionary transition [37].

## A Class IIa aldolase enzyme from Trichoplax functions as a Mud binding protein

To empirically evaluate the *Placozoan* class IIa enzyme and domain genes identified, we began by modeling tetrameric structures of those from a prototype member, *Trichoplax adhaerens* [38]. *Trichoplax* form a ciliated bilayer of cells that perform fundamental coordinated functions such as locomotion and signal transduction [38,39]. As shown in Fig 10A, AF3 predicted assemblies similar to those found for bacterial enzyme and Hts domain aldolase tetramers. The lower ipTM and pTM confidence scores for the domain assembly likely result from 'Low' confidence per-residue plDDT scores found in each protomer CThelix (not depicted). Notably, AF3 modeled $Zn^{2+}$ ions within the conserved histidine triad retained in the *Trichoplax* enzyme sequence and structure, supporting the notion that these retain its aldolase activity (Fig 10A-10B). Similar metal placement was not predicted in the aldolase domain, which has diverged significantly within this catalytic triad (Fig 10B).

We next examined the ability of the *Trichoplax* aldolase proteins to directly bind Mud in pulldown experiments. The *Trichoplax* ALDO$^{DOM}$ bound Mud similarly to Hts, indicating that protein binding is a function present in Adducin proteins in the most basal metazoan (Fig 10C). Surprisingly, we found that the 'full CT' ALDO$^{ENZ}$ from *Trichoplax* bound Mud to an equal extent, and truncation of its CThelix did not improve or impair binding (Fig 10C). Notably, this enzyme, as well as the identified aldolase domain, both contain a partially conserved YLE motif (Fig 10A). We also identified a 'no CT' enzyme in *Trichoplax*, which showed a divergent sequence at the YLE positions and was completely devoid of Mud binding function (Fig 10A, 10C).

Having found that the *Trichoplax* 'full CT' enzyme binds Mud at a level similar to the Hts aldolase domain, we next constructed enzyme chimeras with the *Pseudomonas syringae* aldolase, which is devoid of Mud binding (Fig 4). Specifically,

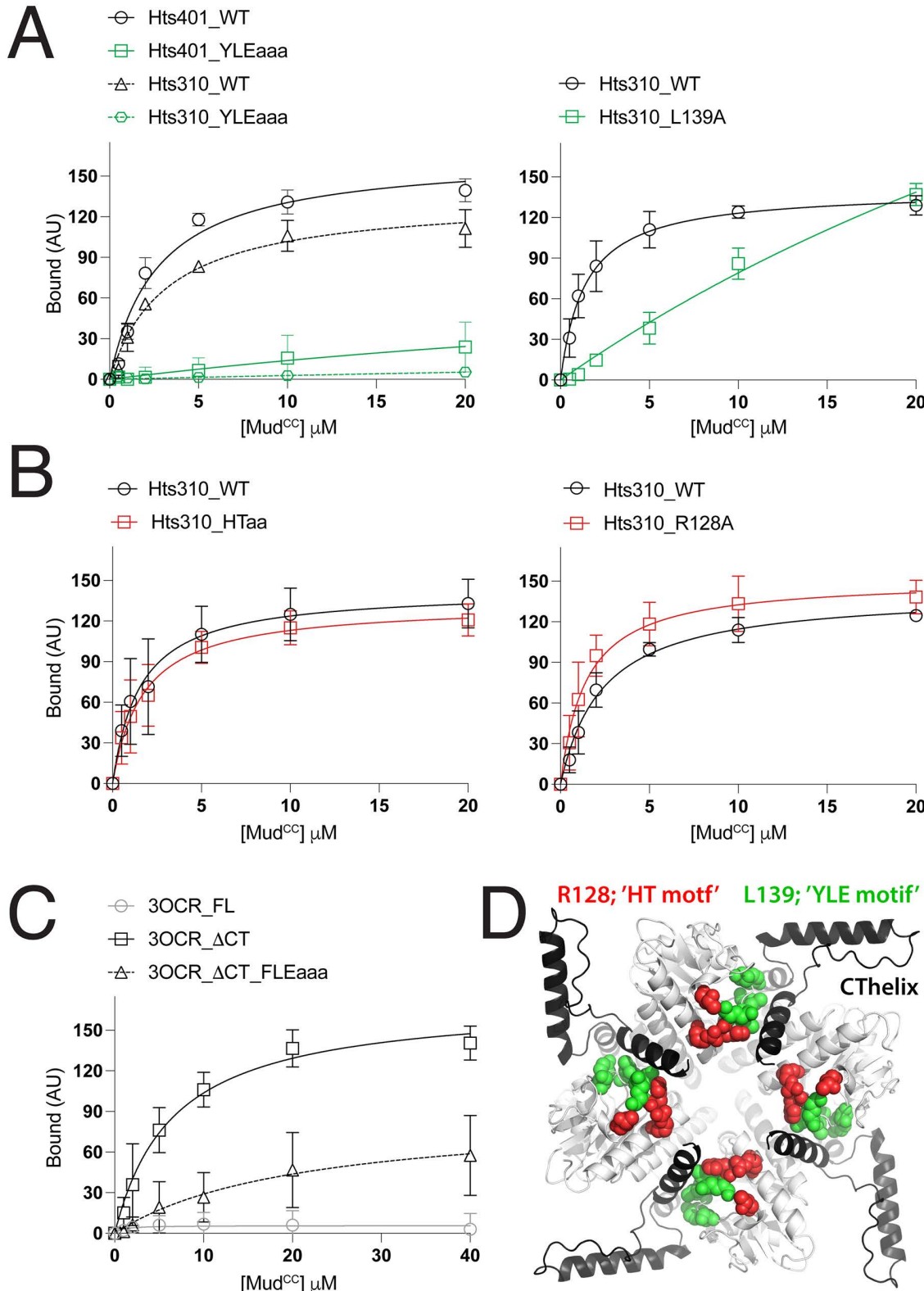

**Fig 7. Mutation of the conserved YLE motif impairs Mud binding. (A):** *Left*: YLE residues were mutated as a triple-alanine mutant (YLEaaa) in either the full-length Hts aldolase domain (Hts401) or the CThelix truncation (Hts310) and tested for Mud binding compared to wild-type (WT). In both cases,

the YLEaaa mutant significantly impaired Mud binding. *Right*: a single alanine mutation at a conserved L139 in the core aldolase domain (Hts310_L139A) also negatively impacted Mud binding dynamics. **(B)**: *Left*: A double alanine mutation at two conserved positions adjacent to the YLE motif was made in the core domain (Hts310_HTaa). This mutant retained full Mud binding function. *Right*: Similarly, a single R128A mutation did not impair Mud binding. **(C)**: Triple-alanine mutation of the FLE motif residues in the *Pseudomonas syringae* aldolase enzyme (3OCR) significantly reduced the gain-of-function Mud binding seen following truncation of the enzyme CThelix. The full-length (FL) enzyme again did not bind Mud. **(D)**: AF3-generated tetrameric assembly of the full-length Hts aldolase domain (*white* with CThelices colored *black*). YLE motif and L139 residues, those whose mutation impaired Mud binding, are shown in *green*, whereas the adjacent residues that were not required for Mud binding are shown in *red*.

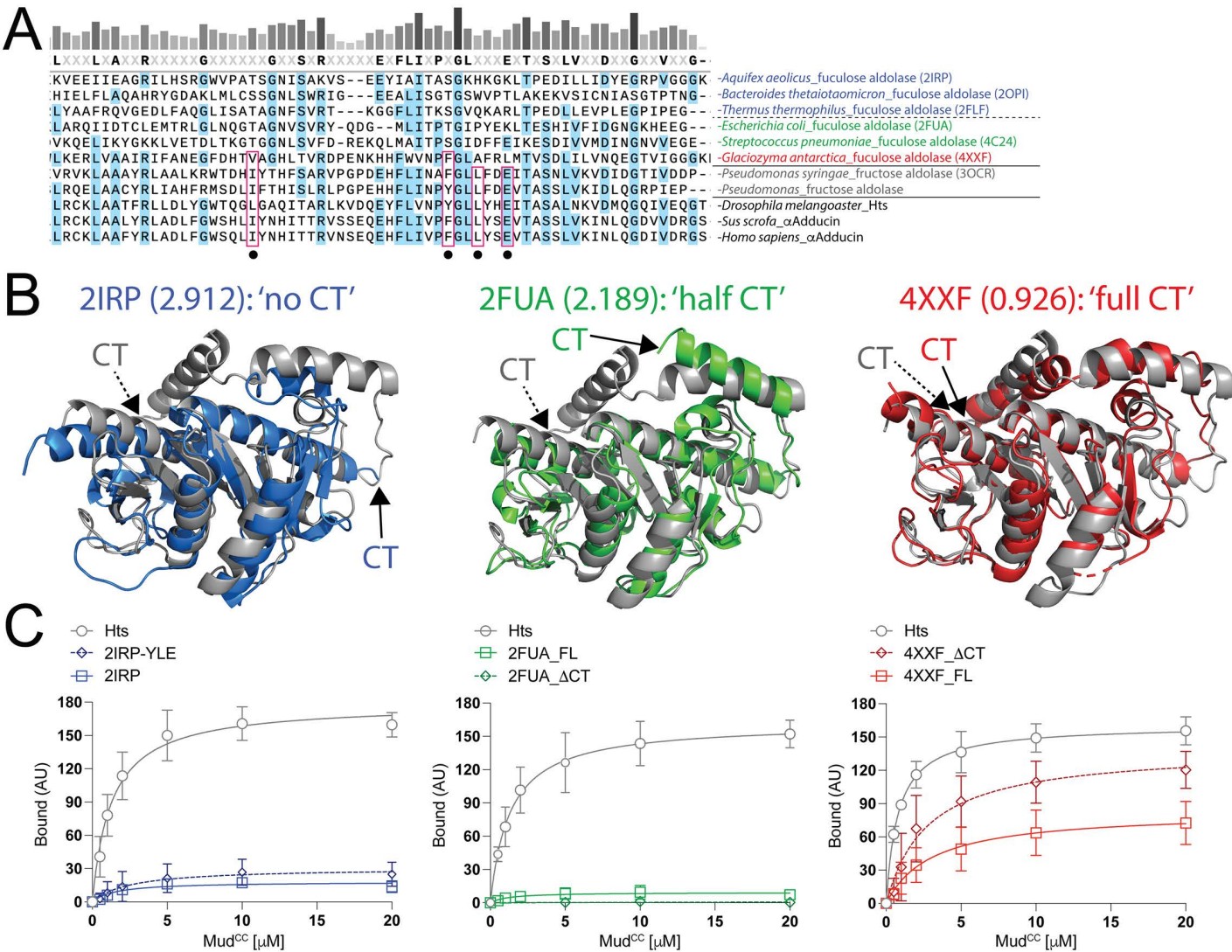

**Fig 8. Divergent aldolase enzymes do not bind Mud despite lack of a complete CThelix. (A):** Multiple sequence alignment among several bacterial aldolase enzymes (with 'aldolase' label and respective PDBid codes where applicable) and aldolase domains (*black* text). Enzyme sequences shown include fuculose aldolases lacking a CThelix (*blue* text), those with a partial CThelix (*green* text), and those with a full CThelix (*red* text). Also shown are the fructose aldolases with full CThelix sequences tested in previous figures (*grey* text). Dots indicate positions corresponding to the YLE motif and L139 in the Hts domain, with those species showing conservation in these positions boxed in *red*. Note that these residues are divergent in fuculose enzymes, particularly those lacking a full CThelix. **(B):** Images depicting structural superimpositions between the *Pseudomonas syringae* fructose aldolase enzyme (3OCR; *grey* in each case) with indicated fuculose aldolases (colors corresponding to that described in panel **A**). RMSD values are shown for each. Each fuculose aldolase enzyme shares structural homology with the fructose aldolase despite significant primary sequence divergence. **(C):** Saturation binding curves for indicated fuculose aldolase enzyme interaction with Mud compared to the Hts aldolase domain. *Left*: the 'no CT' enzyme does not bind Mud despite lack of an intact regulatory CThelix. Mutation of its divergent amino acids to YLE does not confer Mud binding. *Middle*: the 'half CT' enzyme also does not bind Mud, and truncation of its partial CThelix (ΔCT) does not confer binding. *Right*: the 'full CT' enzyme shows modest Mud binding relative to Hts and other fuculose enzymes. This binding is moderately improved upon truncation of the CThelix (ΔCT).

**Fig 9. Sequence database analysis suggest a *Placozoan* origin for aldolase domain proteins. (A):** Structural representations of each subgroup of aldolase proteins are shown. **(B)**: Dots represent the positive identification of an annotated gene for the indicated aldolase protein in BLAST search analysis. For class IIa enzymes, the open dots represent 'no CT', grey dots represent 'half CT', and black dots represent 'full CT' sequences. Higher eukaryotes generally lack class II enzymes but gain expression of aldolase domain proteins (demarked by dashed horizontal line), whereas prokaryotes and lower eukaryotes lack domains and generally show a more diverse enzyme expression profile. Emergence of the aldolase domain-containing Adducin gene family appears similarly in Placozoa and Porifera (*dotted line*), with each also retaining class IIa enzyme genes.

we swapped the CThelix between these enzymes and examined Mud binding ability. Chimeric *Pseudomonas syringae* aldolase that has its CThelix replaced with that from *Trichoplax* was found to have a gain-of-function, showing Mud binding similar to that of the full-length *Trichoplax* enzyme (Fig 10D). In contrast, chimeric *Trichoplax* aldolase with its CThelix replaced with that from *Pseudomonas syringae* was found to have impaired Mud binding relative to the wild-type enzyme (Fig 10D). These data further highlight a regulatory role for the CThelix in protein binding function of the aldolase fold.

### Identification of a conserved Glycine substitution across aldolase domains

Truncation and chimerism of the CThelix, both of which lead to Mud binding in a primitive aldolase enzyme (Figs 4 and 5), represent relatively nontrivial sequence alterations. As such, we sought to identify changes that would have a minimal impact on aldolase enzyme sequence but a similarly large impact on Mud binding function. We considered two criteria to identify possible amino acid substitutions: [1] those located at or near protomer interfaces within the aldolase tetramer, and [2] those with significant distinctions between enzymes and domains conserved across evolution. This analysis revealed one residue in particular, which shows a strong conservation for arginine (or lysine; R/K) across several bacterial aldolase enzymes but that is substituted for glycine (G) in Adducin aldolase

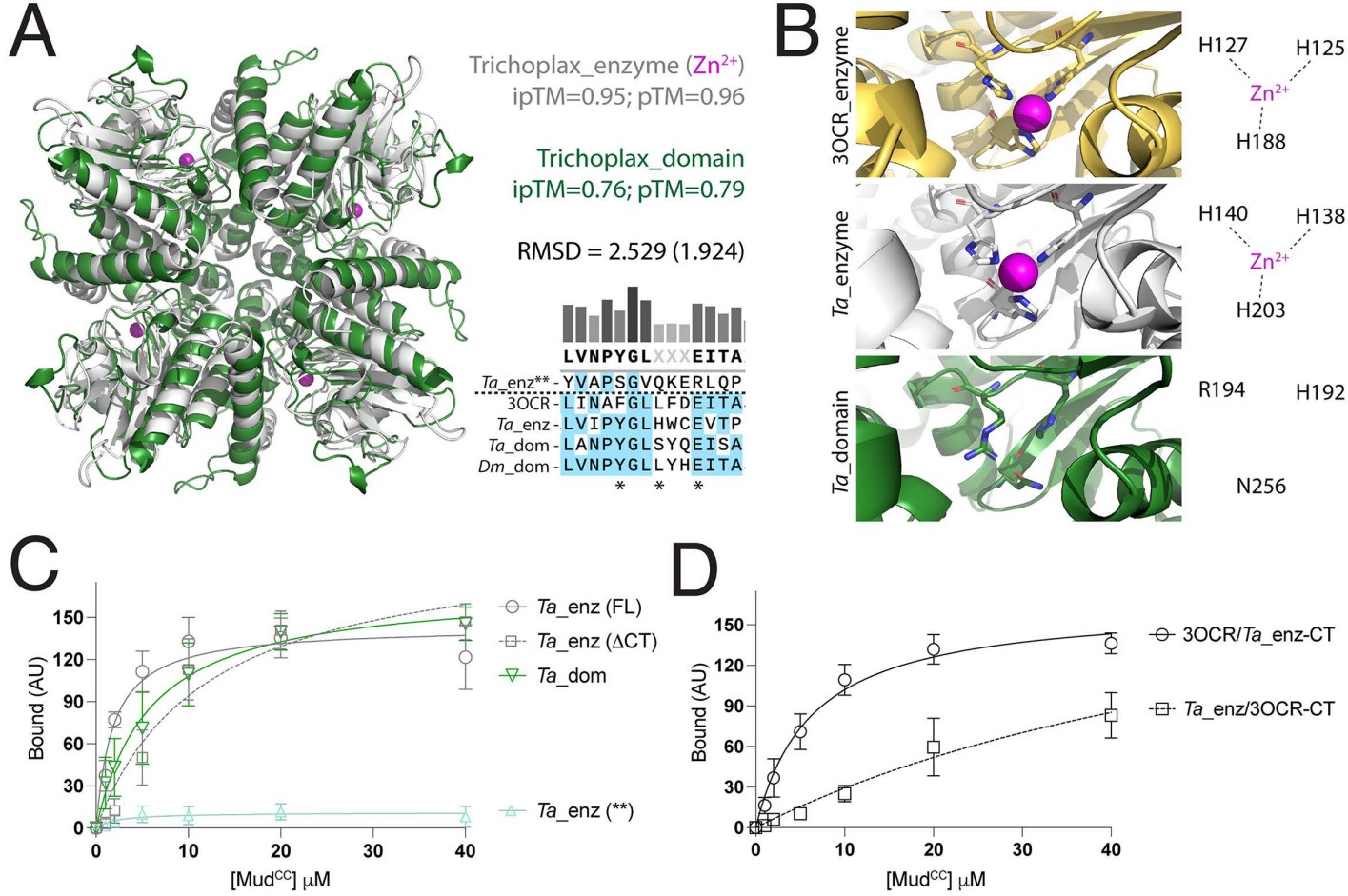

**Fig 10. The *Placozoan* aldolase enzyme has full Mud binding activity and a CThelix with aldolase domain-like regulatory function. (A):** Overlay of AF3-generated tetramer models for the *T. adhearens* (*Ta*) aldolase enzyme (*white*) and Hts-like domain (*green*) shows similarities to other aldolase proteins. AF3 accurately models $Zn^{2+}$ ions (*magenta*) into the conserved histidine triad within the enzyme catalytic site. Multiple sequence alignment shows conservation in the YLE motif (*) in the *Ta* enzyme and domain compared to fly Hts and *Pseudomonas syringae* enzyme (3OCR). Sequence for a 'no CT' fuculose enzyme was also identified in *Ta* (enz**), which shows high sequence divergence in the YLE positions. **(B):** Zoom views of active sites from the *Pseudomonas syringae* aldolase enzyme (3OCR; *top*), *Ta* aldolase enzyme (*middle*), and *Ta* aldolase domain (*bottom*) with the AF3-modeled $Zn^{2+}$ in the *Ta* enzyme coordinated by the histidine triad nearly identical to the bacterial enzyme. In contrast, the *Ta* domain has diverged in two of these histidine residues and does not contain a catalytic metal. **(C):** Saturation binding curves for the *Ta* aldolase proteins. Notably, the full-length (FL) enzyme containing a full CThelix binds Mud equally to the aldolase domain. Truncation of the enzyme CThelix does not improve or impair binding, similar to Hts (see Fig 2). In contrast, the divergent 'no CT' enzyme (**) does not bind Mud. **(D):** Saturation binding curves for chimeric *Trichoplax* and *Pseudomonas syringae* aldolase enzymes are shown. Fusion of the *Trichoplax* CThelix onto the core *Pseudomonas* enzyme (3OCR/*Ta*_enz-CT) shows robust Mud binding compared to the lack of interaction in the full-length bacterial enzyme (see Fig 4). In contrast, fusion of the *Pseudomonas* CThelix onto the core *Trichoplax* enzyme shows a moderate suppression of binding compared to the strong Mud binding seen in full-length *Ta* enzyme (compare to panel **C**).

domains (Fig 11A). Although glycine was also identified in a subset of aldolase enzymes, it was unanimously present across domain sequences analyzed. Substitution of a cationic arginine for glycine represents a significant chemical change, and the strong conservation of these distinct amino acids further suggests a potential functional consequence. In both aldolase enzyme and domain structures, this residue is positioned immediately after the first α-helix of the aldolase fold, which packs against the CThelix (Fig 11B-11C). In the aldolase enzyme, the arginine side chain points towards a central cavity formed in the tetrameric assembly. This results in the guanidinium

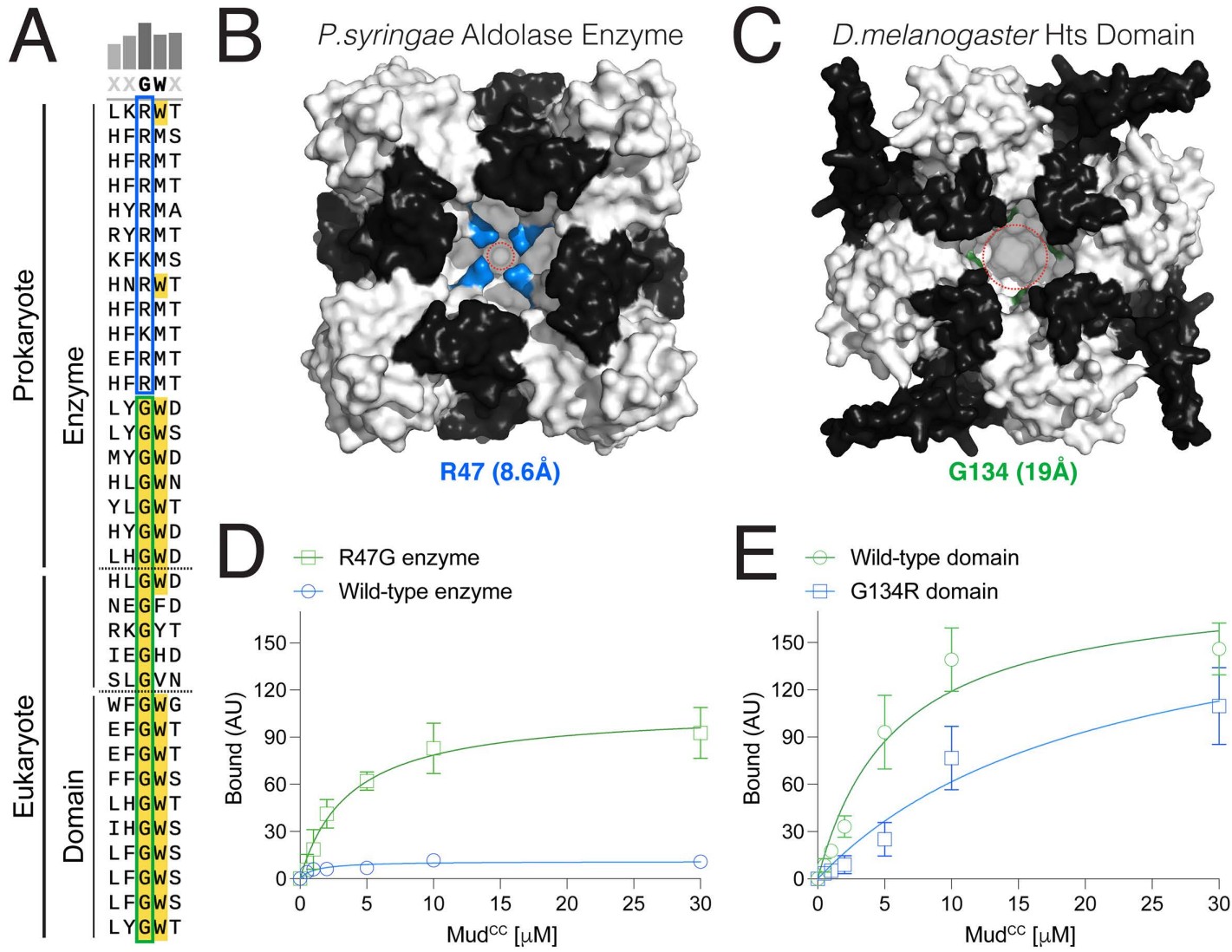

**Fig 11. A single Arginine-to-Glycine substitution permits Mud binding in a bacterial aldolase enzyme. (A):** Multiple sequence alignment of aldolase enzyme and domain sequences demonstrates two distinct subgroups based on the highlighted position. A group of prokaryotic enzymes have a conserved arginine (R) or lysine (K) (*blue box*). Additional prokaryotic enzymes, along with primitive eukaryotic enzymes, have substituted this position with a glycine (G) (*green box*). Aldolase domains also have a strictly conserved glycine in this position. **(B):** Tetrameric assembly of the *P. syringae* aldolase enzyme (RCSBid: 3OCR) is shown in surface rendering with the core fold in *white* and CThelix in *black*. Arginine at position 47 (R47) in each protomer is colored *blue*, the side chains of which point toward a central core of the tetramer to create a cavity of ~8.6Å diameter, denoted by the *dashed red* circle. **(C):** AF3 model for the tetrameric assembly of *D. melanogaster* Hts aldolase domain is shown in surface rendering with the core fold in *white* and CThelix in *black*. Glycine at position 134 (G134) in each protomer is colored *green*. The absence of a non-hydrogen side chain creates a cavity of ~19Å diameter, denoted by the *dashed red* circle. **(D):** Saturation binding curves for Wild-type (*blue*) and R47G (*green*) *P. syringae* aldolase enzymes are shown. The R47G mutant shows significant Mud binding compared to the wild-type enzyme. **(E):** Saturation binding curves for Wild-type (*green*) and G134R (*blue*) Hts aldolase domains are shown. The G134R mutant shows reduced Mud binding affinity compared to the wild-type domain.

R-groups of all four protomer arginine residues in the tetramer facing one another and packing closely together to form a narrow central cavity (8.6Å diameter; Fig 11B). In contrast, the lack of a non-hydrogen side chain in the aldolase domain glycine results in a loss of R-group packing and the formation of a significantly wider central cavity in the AF3 structural model of the Hts tetramer (19Å diameter; Fig 11C).

### Single arginine-to-glycine change in an aldolase enzyme confers Mud binding

To directly examine the role of this R→G change across aldolase proteins, we constructed an R47G mutation in the *Pseudomonas syringae* enzyme and the reciprocal G134R substitution in the Hts domain and measured Mud binding compared to their respective wild-type controls. Mud binding was significantly enhanced in the single R47G aldolase enzyme substitution (Fig 11D). Binding to the R47G enzyme was not as strong as that seen following truncation of the enzyme CThelix, but nevertheless represents a significant gain-of-function. In contrast, Mud binding was reduced in the G134R substituted Hts aldolase domain, although this effect was more modest compared to the gain-of-function seen in the R47G enzyme (Fig 11E). These results agree with those from alterations in the CThelix, with domain-mimicking changes in the aldolase enzyme producing significantly stronger effects than enzyme-mimicking changes in the Hts aldolase domain. Overall, our results spotlight a single amino acid substitution that may represent a significant step in the evolution of aldolase protein function. It should be noted again that although many bacterial enzymes were found to contain an R/K, others were identified with a G at this position. The consequences of this substitution to the normal enzyme function, for example with respect to their catalytic activity, will require further investigation. Deviation from either R/K or G amino acids in this position was rarely seen across aldolase sequences, underscoring a unique functional role.

## Discussion

Complex eukaryotic organisms are reliant on ubiquitous modular protein domains, many of which function as protein interaction platforms that combine to form multidomain scaffolds [40–43]. Deciphering how such domains emerged is critical to understanding their molecular function. Herein, we have explored the molecular determinants for protein binding function in the Hts/Adducin aldolase domain, which shares sequence and structural homology with a subgroup of extant class IIa aldolase enzymes involved in glycolytic metabolism [6,7]. Protein binding in the ALDO$^{DOM}$ follows a dramatic functional transition from a well-defined primitive function (sugar lyase), substantiating it as an excellent case study for investigating neofunctionalization within a eukaryotic protein domain. Divergence at one or more catalytic residues conserved in the class IIa ALDO$^{ENZ}$ explains the loss of enzymatic function in the Adducin family [7]. Sequence changes that uniquely afford protein binding functionality are less clear, however.

Our results suggest that the CThelix plays a critical regulatory role in Mud binding within the aldolase fold. This conclusion is based on several notable observations: (1) the CThelix has undergone sequence and structural divergence between enzymes and domains, (2) the CThelix is dispensable for Mud binding to the Hts domain, but a seemingly domain-specific loop insertion is essential to binding when the CThelix is intact, (3) truncation of the CThelix affords Mud binding to an otherwise binding-deficient aldolase enzyme, and (4) chimeric fusion of CThelix sequences confers their respective regulatory properties on the reciprocal core aldolase proteins. Thus, Mud binding is a property intrinsic to certain aldolase enzymes that appears to be constrained by a noncompliant CThelix. It is also noteworthy that although the *Pseudomonas* enzymes lacked Mud binding (Fig 4), a yeast aldolase enzyme showed moderate binding that was further improved upon CThelix truncation (e.g., 4XXF protein; Fig 8B-8C). Furthermore, the *Placozoan* aldolase enzyme demonstrated robust Mud binding indistinguishable from both *Placozoan* and *Drosophila* Hts domains (Fig 10C). These results are consistent with an evolutionary progression toward protein binding function. Also of note, removal of the large Hts CThelix loop impaired Mud binding, yet neither yeast nor *Placazoan* enzymes contain such an insert and have CThelices structurally more similar to the *Pseudomonas syringae* aldolase (Figs 8B and 10A). Thus, this sequence element does not appear to be required for protein binding, but instead may provide additional regulatory control within the Hts/Adducin domain.

In contrast to this apparent regulatory role for the CThelix, our studies also identify the YLE motif as a putative binding site for Mud that lies within the aldolase core fold. This conclusion is based on several observations as well: (1) mutation of this YLE signature in Hts abolishes Mud binding independent of the CThelix, (2) mutation of the conserved FLE motif impairs the gain-of-function Mud binding seen following CThelix truncation in the *Pseudomonas syringage* aldolase

enzyme, and (3) 'no CT' aldolase enzymes across a range of phyla (from bacteria to *Placozoa*), which do not bind Mud despite their natural lack of a CThelix, are highly divergent in YLE sequence. It is also noteworthy that mutation of surface exposed residues on the aldolase tetramer did not affect Mud binding, consistent with an alternative binding site unique from that identified for the actin barbed end. The YLE motif is located at a tetrameric interface where it directly contacts the CThelix of an adjacent protomer (Figs 6 and 7D), which could explain the regulatory role of the CThelix as controlling exposure of the YLE motif. Such a model would require conformational plasticity in the tetramer as CThelix and Mud binding would be predicted to be mutually exclusive at the YLE motif. Evolutionary changes that impact substrate selection and catalysis by altering conformational dynamics are known for many classes of enzymes [44]. Studies on aldolases have suggested conformational flexibility within the tetrameric assembly, although these have been restricted to class I enzymes [45,46]. To our knowledge, the conformational dynamics in class IIa enzymes and Hts/Adducin aldolase domains remain unexplored. Alternatively, altered oligomerization status might also dictate protein interactions of the aldolase domain. Although the tetramer described herein is considered the predominant form, Adducin has been shown to exist in a minority dimeric form that is in an equilibrium with the majority tetrameric form [47]. Dimeric assembly would leave exposed the putative Mud binding site (e.g., YLE motif) in a single protomer. Future studies will be necessary to resolve these open questions.

Aldolase expression patterns across major phyla reveal several notable conclusions: (1) the appearance of Adducin genes in higher eukaryotes coincides with singular expression of class I enzymes and loss of class II genes that are more often co-identified in lower eukaryotes and prokaryotes, (2) unlike their critical evolutionary position in the emergence of other multidomain scaffold genes, *Choanoflagellates* do not appear to represent a similar role for Adducin-related genes, and (3) instead, the basal metazoan phyla *Placozoa*, *Porifera*, and *Cnidaria*, which emerged 750–800 mya [48], each have Hts/Adducin aldolase domain genes suggesting these non-bilaterian animals represent the first appearance of the Adducin family. In the cases of *Placozoa* and *Porifera*, class IIa enzyme genes were also identified, whereas *Cnidaria* appear to represent the most modern phyla to restrict aldolase enzymes to class I alongside their Adducin gene (Fig 9). The full Mud binding capacity observed in the *Placozoan* aldolase enzyme is striking and is consistent with functional evolution of the aldolase fold. Although its enzymatic activity has not been tested empirically, the histidine triad is conserved in the *Placozoan* aldolase enzyme and is predicted to coordinate a catalytic $Zn^{2+}$ by AF3 modeling (Fig 10B). In contrast, the *Placozoan* aldolase domain has divergent sequence in the catalytic triad and does not bind $Zn^{2+}$ in AF3 modeling. The domain is also a modular component of a larger Hts/Adducin gene, whereas the enzyme is a lone aldolase fold protein.

Our results support a model in which the protein-protein interaction function of the aldolase domain co-opted a binding site involved in the tetrameric assembly established within primitive class IIa enzymes. We hypothesize that this occurred following divergence within the regulatory CThelix, along with additional key residues such as R47, that may have increased flexibility within the aldolase tetramer. As mentioned, this model necessitates a conformational plasticity in the domain tetramer, as Adducin has been shown to exist primarily in this oligomeric state with little contribution of a monomeric form [49]. Interestingly, protein conformational dynamics often correlate with evolvability of new functions [50]. For example, sequence changes leading to relatively subtle conformational changes contribute to evolution of ligand specificity in critical enzyme and receptor groups [51,52]. More dramatic examples include those leading to metamorphic protein evolution of fold switching protein domain [53]. Evolution of the modular pseudokinase domain, which plays key regulatory roles in signal transduction [54], is thought to have occurred through sequence changes that altered conformational flexibility in an ancient protein kinase fold [55]. The gain-of-function Mud binding seen in the R47G bacterial aldolase enzyme spotlights one potentially similar route for such conformational changes. In addition to widening a central cavity within the tetramer core due its simple hydrogen side chain (Fig 11B-11C), this substitution might afford increased conformational dynamics due to the inherent flexibility of glycine positions in the protein backbone. As mentioned above, an alternative model for this aldolase gain-of-function could involve altered dynamics of oligomeric assembly. The Adducin aldolase domain exists in a dimer-tetramer equilibrium [27,47], and the dimeric assembly would leave the putative Mud binding site

unoccupied in one protomer unlike in the tetrameric structure where each aldolase fold interacts with its neighboring sub-unit. Interestingly, studies on diverse protein families have shown that small sequence changes, including single substitutions, can provide an evolutionary path to altered oligomerization with functional consequences [56].

Our results with bacterial aldolase enzymes show striking parallels with a recent example of functional evolution in guanylate kinase (GK), an ancient enzyme involved in nucleotide metabolism that underwent neofunctionalization into a protein binding domain conserved across metazoan phyla [33,57–59]. Unlike aldolase, the GK enzyme functions as a monomer and requires a substantial conformation closing following loading of ATP and GMP substrates to otherwise distant binding sites [59–61]. Neofunctionalization of the GK fold occurred by using the existing GMP substrate binding site within the enyzme as the primary site of protein interaction in GK domains, which required no significant sequence deviation [57]. Protein binding, however, is sterically hindered by the conformational closing seen with GMP binding in the GK fold [59]. To resolve this dilemma, a single S→P amino acid substitution led to restrictions in GK enzyme allostery that produced a productive protein-protein interaction GK domain [57,59]. Specifically, the limited backbone dihedral angles of the substituted proline compared to serine severely restrict flexibility at a critical hinge point in the GK fold, stabilizing an 'open' conformation competent for protein binding at the expense of catalytic activity [59]. The strict R/K→G substitution identified in the aldolase fold in our work here suggests this position may play a similarly critical role in this 'enzyme→domain' functional switch. Although glycine, like serine, also has high dihedral angle flexibility that may facilitate Mud binding function, aldolase exists in an oligomeric form and conformational changes in this assembly are not well appreciated. The R47 and G134 positions each occur at the tetrameric interfaces, which may support an alternative interpretation that R47G gain-of-function could be due to altered oligomerization following the dramatic loss of the extended, highly polar guanidinium side chain of arginine. Indeed, the conspicuous occurrence of arginine at oligomeric interfaces has been previously noted in a diverse subset of protein families [62]. Future studies will be necessary to resolve these open questions and molecular models.

Finally, it is relevant to compare the postulated mode of Mud binding to Hts against that of known Adducin interacting partners. The most well-known and characterized interaction with the Adducin ALDO^DOM is with the actin-spectrin complex. Cryo-EM analysis recently revealed that the tetrameric domain binds the actin barbed end using surface exposed residues, including within the CThelix, along with a more extensive network of residues located in extended sequences not part of the formal aldolase core [8]. These extended regions lie outside the sequence boundaries examined in our studies with Mud and, thus, must not be involved in the Hts/Mud complex. Similarly, truncation of the CThelix did not impair Mud binding, nor did alanine substitutions at conserved surface exposed residues, indicating these are also dispensable. Thus, the mode of ALDO^DOM interaction with Mud appears to be unique compared to F-actin. Adducin has also been shown to localize to spindle poles, independent of actin, through association with Myosin X via the aldolase domain [63]. Notably, Adducin spindle association is regulated by CDK1 phosphorylation at a conserved serine within the CThelix, which is suggested to trigger conformational changes [63]. Whether Myosin X interaction requires an intact CThelix or the conserved YLE motif is currently unknown. Exploring these open questions, along with identifying additional binding partners to the aldolase domain, will be important future steps in further understanding the function of this ancient protein fold.

A limitation of our study is the relatively limited number of aldolase sequences examined. Future studies using additional aldolase sequences from critical evolutionary transitions could potentially offer additional insights into the functional change in this protein fold. Approaches such as ancestral reconstruction could identify common ancestor sequences to the Adducin gene family that would also be important to examine [64]. Another question unresolved in our current work is how sequences changes leading to Mud binding in aldolase enzymes impacts their catalytic function – are protein binding and sugar metabolism mutually exclusive functions in the aldolase fold; or perhaps the gain-of-function protein binding comes at the expense of impaired enzymatic activity? Examination of the class IIa *Trichoplax* enzyme catalytic function compared to bacterial enzymes could help address this question directly. Lastly, as discussed above, biophysical studies on the conformational dynamics of aldolase proteins and their oligomeric assemblies would also offer additional

insights into the structural plasticity of enzyme and domain tetramers and their importance to catalytic and protein binding functions.

## Supporting information

**S1 Fig. Homology modeling shows strong agreement with AF3 in Hts tetramer structure prediction.** A. Overlay of AF3-generated Hts tetramer (*grey core and black CThelices*) and RosettaCM homology model (*green core and blue CThelices*). Strong alignment is seen with all four aldolase domain cores, whereas the CThelices show noticeable divergence. B. Zoom view of chain A-B interface showing similar positioning of the YLE motif, as well as contacting residues in the adjacent CThelix, in AF3 and RosettaCM Hts models. C. Overlay of AF3-generated Hts tetramer (*grey core and black CThelices*) and SWISS-MODEL homology model (*orange core and blue CThelices*). Similar to the RosettaCM model, strong alignment is seen with all four aldolase domain cores, whereas the CThelices again display noticeable divergence. D. Zoom view of chain A-B interface showing similar positioning of the YLE motif, as well as contacting residues in the adjacent CThelix, in AF3 and SWISS-MODEL Hts models.
(TIF)

**S2 Fig. Molecular dynamics (MD) simulations demonstrate stability of the AF3-generated Hts tetramer structure.** A. Overlay between starting AF3 model (*black*) and following replicate 1 of a 200 ns MD simulation (*red*). Asterisk indicates a CThelix showing structural divergence, whereas the core of the tetramer remains stable. B. Overlay between starting AF3 model (*black*) and following replicate 2 of a 200 ns MD simulation (*red*). Asterisk indicates a CThelix showing structural divergence, whereas the core of the tetramer remains stable. C. Overlay between starting AF3 model (*black*) and following replicate 3 of a 200 ns MD simulation (*red*). Asterisk indicates a CThelix showing structural divergence, whereas the core of the tetramer remains stable. D. Plot of 3 simulation average for RSMD changes throughout 200 ns simulation. E. Plot of 3 simulation average for radius of gyration (Rg) changes throughout 200 ns simulation. F. Plot of 3 simulation average for solvent accessible surface area (SASA) changes throughout 200 ns simulation.
(TIF)

**S1 Table. List of sequences used for BLAST-based database searches related to results presented in Fig 9.** Listed are the gene description, organism, and NCBI URL link for specific protein sequences used to identify related sequences across taxa.
(DOCX)

**S1 File. Raw images.** Compilation of all original digital images for SDS-PAGE gels. These images lack any cropping or adjustments that were made in the preparation of final images presented in relevant Figures.
(PDF)

**S2 File. Minimal data set.** Compilation of all original data for measurements of Mud$^{CC}$ binding to listed proteins. These numbers are those used to generate saturation binding curves in relevant Figures. Individual sheets contain data for respective Figures.
(XLSX)

## Author contributions

**Conceptualization:** Christopher A. Johnston.

**Data curation:** Christopher A. Johnston.

**Formal analysis:** Christopher A. Johnston.

**Funding acquisition:** Christopher A. Johnston.

**Investigation:** Marina E. Seheon, Amalia S. Parra, Christopher A. Johnston.

**Methodology:** Christopher A. Johnston.

**Project administration:** Christopher A. Johnston.

**Resources:** Christopher A. Johnston.

**Supervision:** Christopher A. Johnston.

**Validation:** Christopher A. Johnston.

**Writing – original draft:** Christopher A. Johnston.

**Writing – review & editing:** Marina E. Seheon, Amalia S. Parra, Christopher A. Johnston.

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
