## [Decision Letter · Decision Letter 0]

15 Jan 2025

Dear Dr. Johnston,

Thank you for submitting your manuscript to PLOS ONE. After careful consideration, we feel that it has merit but does not fully meet PLOS ONE’s publication criteria as it currently stands. Therefore, we invite you to submit a revised version of the manuscript that addresses the points raised during the review process.

We look forward to receiving your revised manuscript.

Kind regards,

Soumyananda Chakraborti, PhD

Academic Editor

PLOS ONE

Additional Editor Comments:

The manuscript titled "Molecular and evolutionary determinants for protein interaction within a class II aldolase/Adducin domain" investigates the functional evolution of the class II aldolase domain (ALDODOM), examining its transformation from an enzymatic protein to a protein-binding domain. Using AlphaFold-based modeling, biochemical binding assays, and evolutionary sequence analysis, the study focuses on the interaction between the Drosophila ALDODOM and the spindle regulator, Mud, uncovering critical molecular determinants that facilitated this functional transition. While the research is robust and provides significant insights into protein evolution, improvements in organization, clarity, and presentation could enhance the manuscript’s accessibility and impact. Below are my detailed suggestions for refinement:

Major Points for Improvement

Introduction:

• The introduction provides adequate context but would benefit from greater conciseness and improved structure. A streamlined narrative emphasizing the evolutionary and functional transitions of ALDODOM would enhance reader engagement.

Methodology:

• The materials and methods section is comprehensive but could be better organized. Dividing it into distinct subsections with numbered headers for structural modeling, protein purification, and assays would improve readability.

• Include more explicit details on the statistical analysis for biochemical assays, such as error margins, sample sizes, and replicates, to ensure reproducibility and transparency.

Results:

• The results section is thorough but could be presented more clearly by breaking it into smaller subsections with descriptive subheadings (e.g., "Role of the YLE motif," "Impact of CThelix truncation"). This would make key findings more accessible.

• Some figure captions are overly technical and could be simplified while retaining necessary details to aid comprehension.

• Consider reformatting figures with clearer labeling and more distinct color schemes, particularly for structural overlays, to emphasize features discussed in the text.

Discussion:

• The discussion integrates the findings with broader evolutionary implications effectively but could benefit from highlighting unanswered questions and proposing directions for future research.

• Incorporate recent references on advancements in structural modeling tools and studies on evolutionary transitions in protein domains to provide contemporary context.

• Expand the comparative analysis with other evolutionary examples (e.g., guanylate kinase evolution) to draw parallels and contrasts that reinforce the significance of the findings.

Figures and Tables:

• Enhance the clarity of structural overlay figures with improved labeling and distinct colors to differentiate discussed features.

• Add a table summarizing key mutations, their evolutionary significance, and functional impacts to consolidate the findings effectively.

Major Points for Improvement

• Conduct a thorough review to eliminate grammatical errors, typos, and verbose sections.

• Simplify overly technical language to improve readability without compromising scientific accuracy.

By addressing these areas, the manuscript can achieve a more refined and impactful presentation, making it more accessible to a broader audience while maintaining its scientific rigor.

Reviewers' comments:

Reviewer's Responses to Questions

**Comments to the Author**

1. Is the manuscript technically sound, and do the data support the conclusions?

Reviewer #1: Partly

2. Has the statistical analysis been performed appropriately and rigorously?

Reviewer #1: I Don't Know

3. Have the authors made all data underlying the findings in their manuscript fully available?

Reviewer #1: No

4. Is the manuscript presented in an intelligible fashion and written in standard English?

Reviewer #1: Yes

Reviewer #1: The manuscript is interesting. However, I have certain issues:

1. The authors used alfa fold for structure generation. They should try other methods also and compare their results. Since the CryoEM structure of a homologous protein is available, the authors must try homology modelling.

2. The built structure must be validated using MD simulations in triplicate.

3. The binding assay presented here must also be validated by docking simulations using at least 3 different protocols.

**Do you want your identity to be public for this peer review?** For information about this choice, including consent withdrawal, please see our Privacy Policy

Reviewer #1: No

---

## [Author Response · Author response to Decision Letter 1]

28 Apr 2025

Additional Editor Comments:

The manuscript titled "Molecular and evolutionary determinants for protein interaction within a class II aldolase/Adducin domain" investigates the functional evolution of the class II aldolase domain (ALDODOM), examining its transformation from an enzymatic protein to a protein-binding domain. Using AlphaFold-based modeling, biochemical binding assays, and evolutionary sequence analysis, the study focuses on the interaction between the Drosophila ALDODOM and the spindle regulator, Mud, uncovering critical molecular determinants that facilitated this functional transition. While the research is robust and provides significant insights into protein evolution, improvements in organization, clarity, and presentation could enhance the manuscript’s accessibility and impact. Below are my detailed suggestions for refinement:

Major Points for Improvement

Introduction:

• The introduction provides adequate context but would benefit from greater conciseness and improved structure. A streamlined narrative emphasizing the evolutionary and functional transitions of ALDODOM would enhance reader engagement.

To keep the introduction more focused on Aldolase function and evolution, we have moved the discussion of GK evolution to the discussion, which is now found on Lines 583-605.

Methodology:

• The materials and methods section is comprehensive but could be better organized. Dividing it into distinct subsections with numbered headers for structural modeling, protein purification, and assays would improve readability.

We have added numberings (1-5) for subsections in the Methodology section for better clarity.

• Include more explicit details on the statistical analysis for biochemical assays, such as error margins, sample sizes, and replicates, to ensure reproducibility and transparency.

We have added some additional language to help better explain the number of measurements in each experiment and the number of replicates used to generate binding graphs shown in the figures. This can be found on lines 165-167 and 173-174.

Results:

• The results section is thorough but could be presented more clearly by breaking it into smaller subsections with descriptive subheadings (e.g., "Role of the YLE motif," "Impact of CThelix truncation"). This would make key findings more accessible.

We have broken up several larger sections of the Results and added additional subheadings to make them more accessible to the reader as suggested.

• Some figure captions are overly technical and could be simplified while retaining necessary details to aid comprehension.

• We have modified most figure captions in an attempt to simplify their presentation for readers.

• Consider reformatting figures with clearer labeling and more distinct color schemes, particularly for structural overlays, to emphasize features discussed in the text.

• Many of the structural overlays contain many highlighted segments requiring use of unique colors, some of which are less bright than others unfortunately. We considered alternative color schemes but did not feel they improved visibility.

Discussion:

• The discussion integrates the findings with broader evolutionary implications effectively but could benefit from highlighting unanswered questions and proposing directions for future research.

• We have added an additional paragraph at the end of the Discussion section that offer a few additional future research directions that our study supports.

• Incorporate recent references on advancements in structural modeling tools and studies on evolutionary transitions in protein domains to provide contemporary context.

• We have added short discussions of the following references on protein domain evolution. These have been added to the Introduction and/or Discussion sections of the revised manuscript:

o B. K. Muralidhara (2008)

o M. J. Harms (2014)

o A. F. Dishman (2021)

o A. Kwon (2019) and J. Boudeau (2006)

o A.S. Pillai (2022)

o G.K. Hochberg (2017)

• Expand the comparative analysis with other evolutionary examples (e.g., guanylate kinase evolution) to draw parallels and contrasts that reinforce the significance of the findings.

• We have attempted to expand the discussion of GK evolution and provide a better comparison with the present findings as suggested. This includes a more thorough discussion of both the similarities and distinct differences compared to the current findings presented here with aldolase.

Figures and Tables:

• Enhance the clarity of structural overlay figures with improved labeling and distinct colors to differentiate discussed features.

• As mentioned above, we considered alternative color formats but were unsatisfied with the results and did not feel they improved figure presentation.

• Add a table summarizing key mutations, their evolutionary significance, and functional impacts to consolidate the findings effectively.

• We have added ‘Table 1’ with information suggested. We agree that this improves reader access to key findings from the rather extensive list of mutants presented in the manuscript.

Major Points for Improvement

• Conduct a thorough review to eliminate grammatical errors, typos, and verbose sections.

• We have reviewed the manuscript and attempted to correct all grammatical errors.

• Simplify overly technical language to improve readability without compromising scientific accuracy.

By addressing these areas, the manuscript can achieve a more refined and impactful presentation, making it more accessible to a broader audience while maintaining its scientific rigor.

• We have attempted to improve readability to a broad audience by minimizing technical language as requested.

Comments to the Author

Reviewer #1: The manuscript is interesting. However, I have certain issues:

1. The authors used alfa fold for structure generation. They should try other methods also and compare their results. Since the CryoEM structure of a homologous protein is available, the authors must try homology modelling.

• We have completed two homology modeling experiments to complement the AlphaFold-3 structure as suggested. This includes a model based on the CryoEM structure of porcine Adducin heterotetramer using Rosetta Comparative Modeling, along with a second homology model using SWISS-MODEL and the homotetrameric aldolase enzyme crystal structure from Pseudomonas. We appreciate the suggesting and believe this adds additional confidence of the Hts modeling and insights into its structural state. Results are found in Lines 243-251, 360-369, and in Supplemental Figure S1.

2. The built structure must be validated using MD simulations in triplicate.

• We include in the revised manuscript MD simulations as requested. These simulations were done in triplicate, with additional details included in the updated Materials and Methods section (see subsection “5. Molecular Dynamics (MD) simulations:” Lines 190-198). These experiments further validate the tetrameric assembly of the Hts aldolase domain and indicate that it is a stable assembly. Results are found in Lines 252-264 and in Supplemental Figure S2.

3. The binding assay presented here must also be validated by docking simulations using at least 3 different protocols.

• We thank the reviewer for this suggestion but have not included docking simulations in our revised manuscript. The binding assays and protocols used are a standard approach that alone represent a quantitative and empirical assessment of Hts/Mud interactions and effects of various mutations on binding. As such, we believe that our interpretations of the results are not contingent on docking experiments. We hope the reviewer respects our decision on this point.

---

## [Decision Letter · Decision Letter 1]

27 Jul 2025

Dear Dr. Johnston,

Thank you for submitting your manuscript to PLOS ONE. After careful consideration, we feel that it has merit but does not fully meet PLOS ONE’s publication criteria as it currently stands. Therefore, we invite you to submit a revised version of the manuscript that addresses the points raised during the review process.

We look forward to receiving your revised manuscript.

Kind regards,

Bhumi Nath Tripathi

Academic Editor

PLOS ONE

Journal Requirements:

Reviewers' comments:

Reviewer's Responses to Questions

**Comments to the Author**

Reviewer #1: (No Response)

Reviewer #2: (No Response)

2. Is the manuscript technically sound, and do the data support the conclusions?

Reviewer #1: (No Response)

Reviewer #2: Yes

3. Has the statistical analysis been performed appropriately and rigorously?

Reviewer #1: (No Response)

Reviewer #2: N/A

4. Have the authors made all data underlying the findings in their manuscript fully available?

Reviewer #1: (No Response)

Reviewer #2: Yes

5. Is the manuscript presented in an intelligible fashion and written in standard English?

Reviewer #1: (No Response)

Reviewer #2: Yes

Reviewer #1: The authors addressed my comments. However, regarding the docking simulations, the spirit of the comment is to have a consensus decision. I know the protocol used for calculation of binding interaction is a standard one but there is no harm in checking the results with docking.

Reviewer #2: (No Response)

**Do you want your identity to be public for this peer review?** For information about this choice, including consent withdrawal, please see our Privacy Policy

Reviewer #1: No

Reviewer #2: **Yes: ** Sana Fatima Ikram

---

## [Author Response · Author response to Decision Letter 2]

1 Aug 2025

PLOS ONE Reviewer Report

Manuscript Title: Molecular and evolutionary determinants for protein interaction within a class II aldolase/Adducin domain

Recommendation to the Editor: This manuscript is suitable for publication with minor revisions.

We would like to sincerely thank the reviewer for their time and willingness to review our manuscript and offer insightful feedback. We believe we have addressed each of the comments and that our revised manuscript is improved as a result. Below are specific responses to the reviewer’s comments.

Comments:

Lines 34–43: The explanation of neofunctionalization and domain evolution contains redundant wording. Condense this section to enhance readability and focus.

• We have attempted to reword this section to be more concise and focused by removing redundant wording as suggested. We believe the paragraph has improved through this advice. The revised section is now Lines 33-38.

Line 74–77: The sentence “Thus... remains unknown.” is vague. Rephrase it to clearly state the hypothesis being tested.

• We have attempted to rephrase this sentence to be a more clear and direct statement of the study’s goal as suggested. This can be found on Lines 82-85 of the revised manuscript.

Lines 186–211: The long list of BLAST URLs in the Methods section is distracting. Consider relocating URLs to the supplementary data or the data availability section.

• We have moved all URL links to a new Supplementary Table S1 as suggested. We agree this makes this section of the methods much less distracting in its presentation. A reference in the main text is located on Lines 197-199.

Lines 215–259: The section on structural modelling is technically dense and difficult to follow. Add a summarizing sentence at the end to explain how the modelling informed or guided hypothesis generation.

• We have added two sentences at the end of this paragraph. The first adds additional summary of the key interpretation of the modelling, and the second makes clear that these initial results were an important step to the remainder of the study on how Hts and Mud interaction. These revisions can be found on Lines 250-254.

Lines 437–454: The Trichoplax analysis is strong but densely written. Insert a brief sentence explaining why the Placozoa is evolutionarily significant.

• We have added some additional, short descriptions of Placozoa and Trichoplax that provide more perspective on their evolutionary significance, along with additional references for readers. These can be found on Lines 453-456 and 461-462 in the revised manuscript.

Lines 617–621: “through co-option of the enzyme GMP substrate binding site…” is not grammatically incorrect, but it is potentially unclear. Revise it for clarity and grammatical accuracy. When describing domain repurposing or neofunctionalization, clarity is key, especially for enzyme active sites or binding pockets. Therefore, a slight rephrasing will improve reader comprehension without altering the scientific meaning.

• We have made a slight modification to this text to hopefully reduce any confusion among readers as suggested. This new phrasing can be found on Lines 629-631 in the revised manuscript.

---

## [Decision Letter · Decision Letter 2]

28 Oct 2025

Molecular and evolutionary determinants for protein interaction within a class II aldolase/Adducin domain

PONE-D-24-58129R2

Dear Dr. Johnston

We’re pleased to inform you that your manuscript has been judged scientifically suitable for publication and will be formally accepted for publication once it meets all outstanding technical requirements.

Kind regards,

Mohd Akbar Bhat, Ph.D.

Academic Editor

PLOS ONE

Additional Editor Comments (optional):

Reviewers' comments:

Reviewer's Responses to Questions

**Comments to the Author**

Reviewer #2: (No Response)

Reviewer #3: All comments have been addressed

2. Is the manuscript technically sound, and do the data support the conclusions?

Reviewer #2: (No Response)

Reviewer #3: Yes

3. Has the statistical analysis been performed appropriately and rigorously?

Reviewer #2: (No Response)

Reviewer #3: Yes

4. Have the authors made all data underlying the findings in their manuscript fully available?

Reviewer #2: (No Response)

Reviewer #3: Yes

5. Is the manuscript presented in an intelligible fashion and written in standard English?

Reviewer #2: (No Response)

Reviewer #3: Yes

Reviewer #2: (No Response)

Reviewer #3: The authors have addressed all the concerns raised by the reviewers. The manuscript can be accepted for publication.

**Do you want your identity to be public for this peer review?** For information about this choice, including consent withdrawal, please see our Privacy Policy

Reviewer #2: **Yes: ** Sana Fatima Ikram

Reviewer #3: **Yes: ** SHIWALI GOYAL

---

## [Editor Report · Acceptance letter]

PONE-D-24-58129R2

PLOS ONE

Dear Dr. Johnston,

I'm pleased to inform you that your manuscript has been deemed suitable for publication in PLOS ONE. Congratulations! Your manuscript is now being handed over to our production team.

Kind regards,

on behalf of

Dr. Mohd Akbar Bhat

Academic Editor

PLOS ONE